# On the Generalization Superiority of Flat Representation Manifolds for Deep Learning Machines

## Abstract

While modern (deep) Neural Networks (NN) with their high number of parameters have the ability to memorize training data, they achieve surprisingly high accuracies on test sets. One theory that could explain this behavior is based on the manifold hypothesis: real-world high-dimensional input data lies near low-dimensional manifolds. A NN layer transforms the input manifold, arriving at a so-called representation manifold. The NN learns transformations which flatten and disentangle the manifolds layer by layer. In this way, the NNs learn the structure of the data instead of memorizing. Under the manifold hypothesis, we demonstrate that flat manifolds (subsets of affine linear subspaces) in the second-to-last layer of a classification network ensure perfect class separability in the noiseless case. In regression tasks, we derive an upper bound on the generalization error which decreases as the input manifold becomes flatter. In the case of almost flat manifolds, the bound can be modified to be even lower. These results support the argument that flat input manifolds improve generalization. However, we argue that the results can also be used to show that flatter representation manifolds improve generalization. Further, we conduct numerical experiments to show that these findings apply beyond strict theoretical assumptions. Based on our results, we argue that a flatness promoting regularizer, combined with an $L^1$-regularizer, could enhance the generalization of Neural Networks.

## 1 Introduction

Neural Networks (NN) have become a popular tool in the machine learning portfolio for various applications such as image and language processing, signal processing, financial modeling, or as surrogate for physical simulations. The fully connected feed forward network may be one of the most well-known variants, but others have also been developed for specific applications, such as Convolutional Neural Networks for image processing (Zhao et al., 2024) or Recurrent Neural Networks for time series data (Mienye et al., 2024). Recently, the Transformer model (Vaswani et al., 2017) has gained popularity for many applications. The popularity of Deep Neural Networks (DNNs) originates not only from their ability to model complex training data, but also their often outstanding performance on unseen test data. GoogleNet (Szegedy et al., 2014) for example, has circa 6.8 million parameters, was trained on 1.2 million samples from the ImageNet dataset and achieved a test accuracy for image classification of roughly 93%. The generalization gap, estimated by the difference between training and test accuracy, was only 6%. The high test accuracy is surprising, considering NNs are universal approximators, and as such can approximate any Borel measurable function (Hornik et al., 1989). So, networks such as GoogleNet have the ability to easily memorize the training data which would lead to a bad performance on the test set. Instead, high test accuracies can be achieved after gradient-based optimization. Statistical Learning Theory, while being helpful for simple models such as the perceptron (Suykens & Vandewalle, 1999), does not provide a satisfactory explanation either. The upper bounds on the generalization gap developed in this field, such as those based on VC-dimension (Vapnik & Chervonenkis, 2015) or Rademacher complexity, are vacuous for DNNs (Berner et al., 2023). The reason why DNNs generalize well despite the massive amount of parameters is still a mystery. However, one hypothesis is based on manifolds and the manifold hypothesis.

The manifold hypothesis states that real-world high-dimensional data usually are situated on or near

low-dimensional manifolds. Consider, for example, a dataset consisting of images of one object taken from different angles and with different lighting. Each sample, i.e. image, consists of a vector whose dimension is the number of pixels in the image which can be hundreds to thousands. However, the main variations in the data are angles and lighting. The samples are therefore hypothesized to lie on a low dimensional manifold in the high dimensional space. If there are multiple classes in a dataset, we assume that samples of different classes lie on/near different manifolds. The manifold hypothesis is popular in machine learning. For example, the field of manifold learning (Meilă & Zhang, 2024; Ma & Fu, 2012) is based on this hypothesis with the goal of recovering the low-dimensional manifold from samples. If the input data of a neural network lie on a low-dimensional manifold, then we assume the layer's representations also lie on manifolds. This is also a popular assumption (Ansuini et al., 2019; Kaufman & Azencot, 2023). We call manifolds in NN layers representation manifolds.

Based on the manifold hypothesis, Brahma et al. (2016) introduced a possible explanation for the network's behavior of learning data structure instead of memorizing. Their hypothesis states that NN training automatically disentangles manifolds from different classes. Disentanglement here means flattening and separation of manifolds. In this way, the network does not simply memorize the training data, but learns the (manifold) structure of the data, resulting in a small generalization gap. This would also comply with the findings of Zhang et al. (2016) that the structure of the data is essential for estimating the generalization gap. Brahma et al. (2016) empirically showed that the hypothesis is true for multiple networks and datasets. Based on the manifold hypothesis, we aim to improve NN training through encouraging such disentaglement explicitly by adding a measure of flatness as regularizer. These improvements could take the form of better generalization (as popular networks still have generalization gaps of up to 20% (Rohlfs, 2025)) or faster convergence.

In a similar fashion, the popular $L^1$ and $L^2$ regularizors (Phaisangittisagul, 2016; Vidaurre et al., 2013) were developed based on the idea of VC-dimension and Rademacher complexity. These constrain the space of parameters that is searched during training, thereby constraining the ability of the considered model class. They promote smaller weights which lead to more linear models. This should reduce the VC-dimension based upper bound on the generalization gap. We want to develop a regularizer in a similar way, but based on the manifold hypothesis. The present work is the first step in which we present theoretical benefits of encouraging flatness in representation manifolds.

Various researchers have studied properties of representation manifolds (Ansuini et al., 2019; Facco et al., 2017; Kaufman & Azencot, 2023). Ansuini et al. (2019) found a link between the intrinsic dimension of the manifold in the last two layers and the network's performance. Stubbemann et al. (2023) developed a feature selection strategy based on the intrinsic dimension of the manifold. Psenka et al. (2024) developed an autoencoder network whose goal is to linearize the manifold along its layers. However, apart from Labate & Shi (2024), we only found one paper conducting thorough theoretical analysis based on the manifold hypothesis in combination with neural networks: Kiani et al. (2024) showed that learning under input manifolds of bounded curvature is hard. However, with constraints on the volume and curvature, learnability is guaranteed.

The main contributions of this paper are the following:

- A theorem showing that flat class manifolds in the second-to-last layer of a classification network ensure perfect class separability in the noiseless case (Theorem 2).

- A theorem demonstrating that, in the case of regression tasks, having a less curvy input manifold leads to a decreased upper bound on the generalization gap as opposed to having a more curvy input manifold (Theorem 4).

- An extension of Labate et al.'s Theorem (Labate & Shi, 2024) that yields a tighter upper bound on the generalization gap for regression if the input manifold is close to being flat (Corollary 2).

- Empirical validation of the benefits of flat input manifolds in regression through an experiment whose setup is closer to practical applications of neural networks than the assumptions of Theorem 4 and Corollary 2. (Section 4.2).

This work is structured as follows: In Section 2, we introduce the considered flatness measures and show that flatness is equivalent to the manifold being a subset of an affine linear subspace. In Section 3, we consider classification and present a proof that flat representation manifolds in the second-to-last layer of a classification network guarantee separability, i.e. perfect classification. In Section 4, we consider the case of regression. We show that there is an upper bound on the generalization gap, which decreases with increasing flatness of the input manifold (i.e. manifold of the input data). Further, we show that especially small upper bounds can be achieved if the input manifolds are close to being flat. These findings hold for a certain architecture of neural network. Numerically, we show that flatter input manifolds can also be beneficial in practical settings that differ from this architecture. In Section 5 and 6, we summarize the findings and present next steps and further research directions.

## 2 Curvature of Manifolds

A Riemannian manifold is a smooth manifold equipped with a Riemannian metric. The curviness of such a manifold can be either intrinsic or extrinsic. Intrinsic curvature is inherent to the manifold and is defined by the Riemannian curvature tensor. We are interested in the extrinsic curvature which is determined by the embedding and can be measured by the Second Fundamental Form. As an example, consider one period of a sine-curve as a 1-dimensional manifold embedded into 3-dimensional space. If you decrease the frequency, the extrinsic curvature decreases since the manifold is now more stretched. The intrinsic curvature, however, considers the curvature in the manifold-inherent coordinate system which is embedding-invariant. For an introduction to differential geometry, we refer the reader to Robbin & Salamon (2022).

**Definition 1.** (Robbin & Salamon, 2022) Let $\mathcal{M} \subseteq \mathbb{R}^D$ be a simply connected (Krantz, 1999) $d$-dimensional Riemannian manifold. Let $T_p\mathcal{M}$ be the tangent space at point $p \in \mathcal{M}$. Then, there exists an orthogonal projection $\Pi(p) \in \mathbb{R}^{D \times D}$ which projects points $v \in \mathbb{R}^D$ onto the tangent space $\Pi(p)v \in T_p\mathcal{M}$. The symmetric bilinear map

$$II_p : T_p\mathcal{M} \times T_p\mathcal{M} \to T_p\mathcal{M}^\perp$$
$$II_p(v, w) = (d\Pi(p)w)v$$

is the **Second Fundamental Form** (SFF) on $\mathcal{M}$ in $p \in \mathcal{M}$.
Here,

$$d\Pi(p)w := \frac{d}{dt}\Pi(\gamma(t))|_{t=0},$$

where $\gamma : \mathbb{R} \to \mathcal{M}$ with $\gamma(0) = p$ and $\dot{\gamma}(0) = w$.

**Theorem 1.** *Let $\mathcal{M} \subseteq \mathbb{R}^D$ be a $d$-dimensional simply connected Riemannian manifold. Then, the following are equivalent:*

- *(a) There is a convex set $X \subset \mathbb{R}^d$, a matrix $B \in \mathbb{R}^{D \times d}$ and a vector $b \in \mathbb{R}^D$ so that $\mathcal{M} = \{Bx + b : x \in X\}$.*

- *(b) The manifold $\mathcal{M}$ is flat in the sense that the SFF vanishes $II_p(v, w) = 0 \ \forall p \in \mathcal{M}, \ \forall v, w \in T_p\mathcal{M}$.*

*Proof.* The proof will be presented in the Appendix A.1. $\qquad\square$

Theorem 1 shows that a flat manifold is a subset of an affine linear subspace.
Apart from the SFF, the reach is another tool to characterize the extrinsic curvature of a manifold:

**Definition 2.** For a manifold $\mathcal{M} \subseteq \mathbb{R}^D$, the **reach** is defined as

$$\tau := \max\{r \in \mathbb{R}_{>0}| \ (\forall x \in \mathbb{R}^D \text{ with } d(x, \mathcal{M}) < r : x \text{ has a unique nearest point on } \mathcal{M})\}.$$

where $d(x, \mathcal{M})$ is the euclidean distance from point $x \in \mathbb{R}^D$ to the nearest point on the manifold.

**Remark 1.** In contrast to the SFF, the reach does not depend on the considered point, but is a global property of the manifold. It can be seen as a worst-case measure that considers how much the manifold folds in onto itself. A manifold with low reach either has a section with "tight curls" or is close to self-intersection at some point. It can be shown (Aamari et al., 2019) that the reach is the minimum of

1. the smallest bottleneck of the considered manifold and

2. the minimal radius of curvature, which can be expressed by the SFF.

The reach of a linear subspace, which is a flat manifold with respect to the Second Fundamental Form, is infinite.

## 3 Flat representation manifolds are beneficial for classification tasks

In the case of classification, the manifold hypothesis states that samples of different classes are situated on different manifolds. The following theorem formalizes why flat manifolds in the second-to-last layer of classification networks are beneficial:

**Theorem 2.** *Any two flat manifolds as characterized in Theorem 1 (a) where X is compact, that do not intersect can be separated by a linear Neural Network layer.*
*In other words, for the two hyperplanes $\mathcal{M}_1$ and $\mathcal{M}_2$, there exists a weight matrix $W \in \mathbb{R}^{2 \times D}$ and a bias vector $b \in \mathbb{R}^2$ so*

$$\arg\max(W\tilde{x} + b) = \begin{cases} 0 & \tilde{x} \in \mathcal{M}_1 \\ 1 & \tilde{x} \in \mathcal{M}_2. \end{cases}$$

*The operator $\arg\max$ maps a vector to the index with maximum entry. If multiple entries attain the maximum, the first such index is returned.*

*Proof.* A linear NN layer simply defines a hyperplane. All points on one side of the hyperplane are classified as 0 and all other points as 1. We have to show that compact subsets of affine linear subspaces can be separated by a hyperplane if they do not intersect. The hyperplane separation theorem (Boyd & Vandenberghe, 2004) states that two closed disjoint nonempty convex subsets of $\mathbb{R}^n$ can be strictly separated by a hyperplane if one of the subsets is compact. Note that the considered subspaces are by assumption compact and, due to them being simply connected subsets of affine linear subspaces, they are convex. By applying the hyperplane separation theorem, we conclude the proof. □

**Remark 2.** Theorem 2 shows that flatness of manifolds (as characterized in Theorem 1) is a sufficient condition for separability, but not a necessary one. Figure 1 shows this using examples. For classification, the overall curviness in combination with the space separating the hyperplanes determines whether the classification is successful, rather than smaller, local areas of higher curvature. We will see in the next section that this is different in the regression case.

**Remark 3.** Note that Berner et al. (2023) argued in their Section 4.1 why NNs can, in the presence of a data manifold, reduce the problem to the underlying low-dimensional problem. This is done by partitioning the manifold into suitable neighborhoods and approximating the coordinate charts of the manifold via NNs using the universal approximation property. This means that NNs can recover the low-dimensional manifold-inherent coordinate system. If they can perform this, they can also project the resulting low-dimensional manifold into high-dimensional space, resulting in flat manifolds.

Theorem 2 does not state how to achieve flat manifolds in the second-to-last layer. One idea that we want to consider in future research is to encourage flat representation manifolds by explicitly measuring the flatness and adding this as penalty term in the loss function, resulting in a flatness regularizer.

Further, from Theorem 2 it is not clear whether flatness of class manifolds in all layers is beneficial or only in the last. Frosst et al. (2019) showed empirically that class entanglement in earlier layers can be beneficial for generalization. Kaufman & Azencot (2023) showed that in successful image classification networks, representation manifolds are especially curvy. However, they did not consider different classes separately. Whether encouraging flatness in earlier layers benefits generalization has yet to be studied.

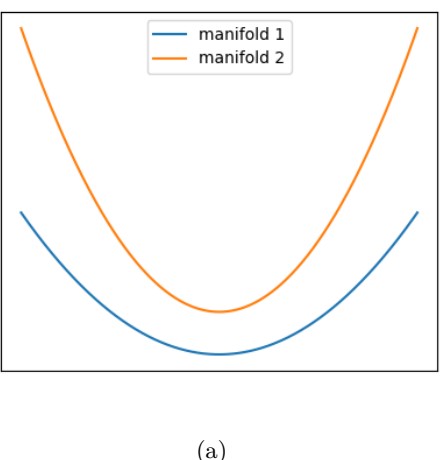
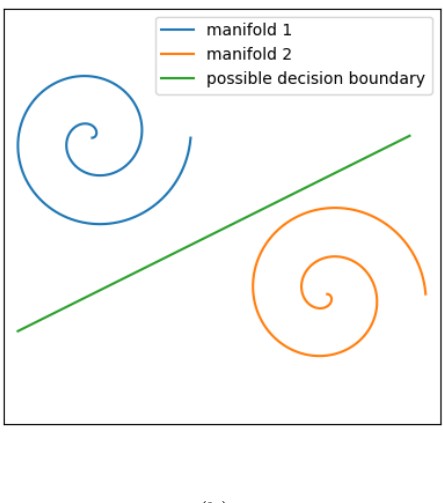

(a)                                                                                            (b)

Figure 1: Two configurations of curvy manifolds that show that flatness is sufficient (a), but not necessary (b) for separability. On the right, one possible separating hyperplane that can be defined by a linear NN layer is displayed.

## 4   Flat input manifolds are beneficial for regression tasks

In this section, we show how flat input manifolds can be beneficial for regression. In Section 4.1, we present a theorem by Labate & Shi (2024) as well as an extension which states that flat manifolds are beneficial for networks' performance. In Section 4.2 we give numerical examples which are closer to real-world applications than the presented theorems.

### 4.1   Theoretical Considerations

We consider the cases of non-flat and flat manifolds separately. We find that the upper bound on the network's generalization gap is lower if the input manifold is flat. We study how far the manifold can deviate from being perfectly flat while still retaining the tighter bound.
In the following, $||.|| := ||.||_2$ is the euclidean norm either in vector or matrix form.

### 4.1.1   Non-flat Manifolds

We first present the Theorem by Labate & Shi (2024). This has several assumptions. For a more thorough explanation and derivation, refer to the original work.
**Assumptions:**

A 1  Observations $\{x_i, y_i\}_{i=1}^n \subseteq \mathcal{M} \times \mathbb{R}$ with $n \in \mathbb{N}$ are available where $\mathcal{M} \subseteq [0,1]^D$ is a $d$-dimensional Riemannian manifold. The samples $x_i$ are drawn independently and identically distributed (i.i.d.) from a random variable with probability density function $\mu$.

A 2  The manifold $\mathcal{M}$ is a compact, smooth Riemannian manifold with $0 \in \mathcal{M}$. Its reach is given by $\tau$.

A 3  The ground truth is $y_i = f_0(x_i) + \epsilon$, where $\epsilon \sim \mathcal{N}(0, \sigma^2)$ is a normally distributed (measurement) error.

A 4 $f_0 \in \mathcal{H}(\beta, \mathcal{M}, R)$ where $\mathcal{H}$ is the ball with radius $R > 0$ in the Hölder space $\mathcal{H}(\beta, \mathcal{M})$ with $\beta \in (0, 1]$. This means

$$f_0 \in \mathcal{H}(\beta, \mathcal{M}, R) = \{f \in C^0(\mathcal{M}) \mid \sup_{x \in \mathcal{M}} f(x) < R \text{ and}$$

$$\sup_{x \neq y \in \mathcal{M}} \frac{|f(x) - f(y)|}{||x - y||^\beta} < R\}.$$

For $\beta = 1$, the second condition is the global Lipschitz-continuity condition.

A 5 The considered machine learning model consists of a projection $\Psi$ followed by a ReLU neural network. This means that the output of the projection is the input for the ReLU NN further detailed in Assumption A 6. The function $\Psi$ has the following form:

$$\Psi : \mathcal{M} \to \mathbb{R}^{d_e}$$
$$\Psi(x) = Ax + c$$

for a matrix $A \in \mathbb{R}^{d_e \times D}$ and a vector $c \in \mathbb{R}^{d_e}$. The scalar $d_e \in \mathbb{N}$ is called effective dimension. The effective dimension $d_e$ together with $A$ and $c$ are chosen in such a way that he projection fulfills the following properties for $\epsilon \in \left(0, \frac{1}{3}\right]$ and $x_1, x_2 \in \mathcal{M}$ with at least some probability $\nu \in (0, 1]$:

$$(1 - \epsilon)||x_1 - x_2|| \leq ||Ax_1 - Ax_2|| \leq (1 + \epsilon)||x_1 - x_2|| \text{ and}$$
$$\Psi(\mathcal{M}) \subseteq [0, 1]^{d_e}.$$

Labate & Shi (2024) show in their Theorem 4 (which we will call Theorem L4 to avoid confusion with Theorem 4 in this work) that this assumption is e.g. fulfilled if the following assumptions are met:

(a) $v_{\mathcal{M}} \geq \left(\frac{21}{2\sqrt{d}}\tau\right)^d$, where $v_{\mathcal{M}}$ is the volume of the manifold and $d$ is the intrinsic dimension of the manifold as introduced in Assumption A 1. Note this assumption is not applicable for flat, compact manifolds where $\tau = \infty$, and therefore $v_{\mathcal{M}}$ would have to be infinite.

(b) The projection function $\Psi$ has the following form

$$\Psi(x) := \frac{1}{4\text{diam}(\mathcal{M})}(Ax - y_0)$$

The matrix $A \in \mathbb{R}^{d_e \times D}$ consists of i.i.d random variables distributed according to $\mathcal{N}(0, 1/d_e)$ and $y_0 \in \mathbb{R}^{d_e}$ ensures that $\Psi(x) \in [0, 1]^{d_e}$ with high probability. For a more detailed derivation of why such a $y_0$ exists, refer to the proof of Theorem 1 in Labate et al.'s original work (Labate & Shi, 2024).

The operator $\text{diam}(\mathcal{M})$ is the diameter of the manifold. Note that the diameter is the maximal geodesic distance between two points on the manifold. The geodesic distance between two points on a manifold is the length of the shortest path between those points, but the path has to be on the manifold. If the manifold is not flat, this is different from the euclidean distance between two points.

The effective dimension in this case is given by

$$d_e = \left\lceil 828 \left( 24d + 2d \log \left( \frac{9\sqrt{d}}{\tau} \right) + \log(2v_{\mathcal{M}}^2) \right) \right\rceil.$$

Here, log denotes the natural logarithm. If the reach $\tau$ is large, i.e. $\mathcal{M}$ has small curvatures and is far apart from self-intersecting, then $d_e$ is small.

A 6 The ReLU network parameters are the solution of the least squares problem

$$\hat{f} = \underset{f \in \mathcal{F}(N,L,B)}{\arg\min} \sum_{i=1}^{n} (f(x_i) - y_i)^2$$

where $\mathcal{F}(N, L, B)$ is the set of all ReLU networks with the specified depth $L \in \mathbb{N}_{>0}$, maximal width $N \in \mathbb{N}_{>0}$ and maximal norm of the weights $B \in \mathbb{R}$. A ReLU network is a fully connected feed forward NN where the activation function is the well-known rectified linear unit function. The parameters $L$, $N$ and $B$ of the neural network depend on $d_e$ as well as some other quantities.

We now present the theorem by Labate et al. Their aim was to show that the network's performance in terms of function approximation based on samples does not depend on the ambient dimension $D$, but only on the intrinsic dimension $d$ of the input manifold $\mathcal{M}$.

**Theorem 3.** *(Labate & Shi, 2024) If Assumptions A 1-A 6 with stricter assumptions A 5a and A 5b are fulfilled, then with probability at least $1 - 2\exp(-n^{\frac{d_e}{2\beta + d_e}})$ for any $n \geq \tilde{N}$ with $\tilde{N}$ sufficiently large it holds that*

$$||\hat{f} - f_0||^2_{L^2(\mathcal{M},\mu)} \leq c_1 n^{-2\beta/(2\beta + d_e)}$$

$$+ \frac{c_2}{n} \left( \log \left( \frac{c_3}{c_4} n^{\beta/(2\beta + d_e)} \right) + 1 \right)^2$$

$$+ \frac{c_5}{n} \log(c_6 n^{\beta/(2\beta + d_e)})$$

$$+ c_7 n^{-\beta/(2\beta + d_e)} \tag{1}$$

$$\leq C n^{-2\beta/(2\beta + d_e)} (1 + \log n)^2 \tag{2}$$

*where $c_i > 0$ are independent of $n$, $i \in \{1, 2, 3, 4, 5, 6, 7\}$ and $C$ depends on $\sigma$, $\beta$, $d_e$, $R$ and the diameter of the manifold $\mathcal{M}$.*

We now present a theorem, which shows that flatter input manifolds decrease the upper bound presented in Theorem 3.

**Theorem 4.** *Let $\epsilon > 0$. If Assumptions A 1-A 6 and Assumptions A 5a and A 5b hold, then there exists an $\hat{N} \in \mathbb{N}$ so that for all $n \geq \hat{N}$ with probability at least $1 - \epsilon$, the upper bound on the generalization gap $||\hat{f} - f_0||^2_{L^2(\mathcal{M},\mu)}$ decreases with decreasing $d_e$.*
*A necessary condition for $d_e$ decreasing when going from manifold $\mathcal{M}_1$ to $\mathcal{M}_2$ is:*

$$\Delta v_{\mathcal{M}} < \frac{v_{\mathcal{M}_1} d}{\tau_1} \Delta \tau$$

*for the change in volume $\Delta v_{\mathcal{M}} = v_{\mathcal{M}_2} - v_{\mathcal{M}_1}$ and the change in reach $\Delta \tau = \tau_2 - \tau_1$ for small enough $\Delta \tau$. This includes the case were $\Delta v_{\mathcal{M}} = 0$ and $\Delta \tau > 0$, i.e. the volume of the manifold does not change while the manifold becomes less curvy.*

*Proof.* For the first statement of the proof, i.e. that the upper bound on the generalization gap decreases with decreasing $d_e$, we present two proofs in the Appendix A.2. The first proof is simpler, but considers an upper bound which is not tight to Equation (1). The second proof is more elaborate, but considers a tighter bound. Here, Equation (1) is only upper bounded to achieve a differentiable function. For further details, we refer the reader to the Appendix A.2.

Consider the second statement, i.e. $d_e$ can only decrease if $\Delta v_{\mathcal{M}} < \frac{v_{\mathcal{M}_1} d}{\tau_1} \Delta \tau$ for small enough $\Delta \tau$. We consider $\tilde{d}_e$, which is $d_e$ from Assumption A 5b, but without the ceiling operation.

$$\tilde{d}_e := 828 \left( 24d + 2d \log \left( \frac{9\sqrt{d}}{\tau} \right) + \log(2v_{\mathcal{M}}^2) \right)$$

$$= k + \delta$$

for an integer $k \in \mathbb{N}$ and $0 < \delta < 1$. This means that $d_e$ only needs a negative change of magnitude $\delta$ to decrease to the next smaller integer. Then, using Taylor expansion, we get

$$\tilde{d}_e(\tau_1 + \Delta\tau, v_{\mathcal{M}_1} + \Delta v_{\mathcal{M}}) = \tilde{d}_e(\tau_1, v_{\mathcal{M}_1}) + 828\left(-\frac{2d}{\tau_1}\Delta\tau + \frac{2}{v_{\mathcal{M}_1}}\Delta v_{\mathcal{M}}\right) + \mathcal{O}(||[\Delta\tau, \Delta v_{\mathcal{M}}]||^2).$$

The new $\tilde{d}_e(\tau_2, v_{\mathcal{M}_2}) = \tilde{d}_e(\tau_1 + \Delta\tau, v_{\mathcal{M}_1} + \Delta v_{\mathcal{M}})$ is smaller than $\tilde{d}_e(\tau_1, v_{\mathcal{M}_1})$ if

$$\left(-\frac{2d}{\tau_1}\Delta\tau + \frac{2}{v_{\mathcal{M}_1}}\Delta v_{\mathcal{M}}\right) < 0$$

$$\Leftrightarrow \Delta v_{\mathcal{M}} < \frac{v_{\mathcal{M}_1}}{\tau_1}d\Delta\tau$$

for small enough $\Delta\tau$ (so the residual term $\mathcal{O}(||[\Delta\tau, \Delta v_{\mathcal{M}}]||^2)$ is negligible). If now the decrease is larger than $\delta$, $d_e$ decreases at least onto the next smaller integer. For this to happen, we need

$$||\delta|| < \left\|828\left(-\frac{2d}{\tau_1}\Delta\tau + \frac{2}{v_{\mathcal{M}_1}}\Delta v_{\mathcal{M}}\right) + \mathcal{O}(||[\Delta\tau, \Delta v_{\mathcal{M}}]^2||)\right\|.$$

$\square$

**Example 1.** A manifold which changes its reach but does not change its volume fulfills one of the condition for $d_e$ decreasing in Theorem 4. This happens, for example, if you imagine unfolding a manifold shaped like a scroll. To formalize this idea, consider the following manifolds where $\tilde{\kappa} > \kappa$:

$$\mathcal{M}_1 = \{[\kappa\cos(\theta), \kappa\sin(\theta), \underbrace{0, ..., 0}_{D-2}]^T : \theta \in [0, \pi]\} \subset \mathbb{R}^D,$$

$$\mathcal{M}_2 = \{[\tilde{\kappa}\cos(\theta), \tilde{\kappa}\sin(\theta), \underbrace{0, ..., 0}_{D-2}]^T : \theta \in [0, \frac{\kappa}{\tilde{\kappa}}\pi]\} \subset \mathbb{R}^D,$$

$$\mathcal{M}_3 = \{[\theta, \underbrace{0, ..., 0}_{D-1}]^T : \theta \in [0, \kappa\pi]\}.$$

All three are 1-dimensional manifolds (i.e. lines) embedded in $\mathbb{R}^D$. $\mathcal{M}_1$ describes a half circle, $\mathcal{M}_2$ with $\tilde{\kappa} > \kappa$ a 'smaller fraction' of a circle and $\mathcal{M}_3$ is a straight line. The volume and diameter of all three manifolds is $\kappa\pi$, while the reach is $\kappa$, $\tilde{\kappa}$ and infinity respectively. Therefore, $\Delta v_{\mathcal{M}} = 0$ while $\Delta\tau_{2,1} = \tau_2 - \tau_1 = \tilde{\kappa} - \kappa > 0$ and $\tau_{3,2} = \infty$. This is an example for manifolds which fulfill the condition of Theorem 4 for a decreasing effective dimension $d_e$.

**Example 2.** In this example we show that the condition in Theorem 4 for decreasing $d_e$ is not always fulfilled. Consider the spherical cap, which is a sphere cut off by a plane. This is our 2-dimensional manifold embedded in 3 dimensions, so $d = 2$ and $D = 3$. Let the sphere have radius $\kappa$ and a cut-off-angle of $\theta$. The volume is $v_{\mathcal{M}} = 2\pi\kappa^2(1 - \cos(\theta))$ and the reach is equal to the radius $\kappa$ as in the previous example, so $\tau = \kappa$. We choose the following values for manifolds $\mathcal{M}_1$ and $\mathcal{M}_2$:

$$\kappa_1 = 0.5, \theta_1 = 0.1,$$
$$\kappa_2 = 1, \theta_2 = 0.2.$$

By choosing a larger radius $\kappa_2$, we make $\mathcal{M}_2$ less curvy. At the same time, the increase in cutoff-angle $\theta_2$ increases the volume of $\mathcal{M}_2$ in relation to $\mathcal{M}_1$. This results in

$$\Delta v_{\mathcal{M}} = v_{\mathcal{M}_2} - v_{\mathcal{M}_1} = 0.11 > 0.016 = \frac{0.008 \cdot 2 \cdot \Delta\kappa}{\kappa_1} = \frac{v_{\mathcal{M}}d}{\tau_1}\Delta\tau.$$

The effective dimensions are

$$d_{e_1} = 43012$$
$$d_{e_2} = 45303.$$

This example shows that the condition described in Theorem 4 for decreasing $d_e$ does not hold in all cases.

**Remark 4.** Labate & Shi (2024) note that for $\mathcal{M}$ being a unit sphere, $d_e \approx 19872d$. However, a more general statement can be made about $d_e$: Inserting Assumption A 5a, $v_{\mathcal{M}} \geq \left(\frac{21}{2\sqrt{d}}\tau\right)^d$, into the definition of $d_e$ (Assumption A 5b) yields that $d_e \geq 27987$, making the bound in Theorem 3 too large to have any meaning in practice.

**Remark 5.** The reason why a larger reach decreases the bound is because if the manifold is less complex (larger reach, lower curvature), the effective dimension $d_e$ (which is the dimension after the projection) can be lower without losing important information during the projection (refer to Assumption A 5b). The reader might notice that the Johnson-Lindenstrauss lemma (Larsen & Nelson, 2017) contains a similar statement to the one in Theorem L4 (Theorem 4 in Labate & Shi, 2024) (refer to Assumption A 5). However, in contrast to the lemma, in Theorem L4 the number of samples has no impact on the dimension of the projection space $d_e$. This means that the statement generalizes to out-of-sample data.

**Remark 6.** Theorem 4 can be applied not only to input manifolds, but also to representation manifolds in the following way: In a NN with $k$ layers, the first layer's representation acts as input for the NN consisting of layers 2 to $k$. Likewise, the second layer's representation acts as input for the NN consisting of layers 3 to $k$, and so on. Therefore, promoting flatness in the layer's representations leads to better performance of the following sub-network, yielding better overall performance.

Recall that the reason for the improved performance is that less complex structures can be represented using less neurons (refer to Remark 5). So, by encouraging the representation manifolds to be flat, we can decrease the width of successive layers without loss of information, thereby retaining good fit to training data and achieving better generalization.

Even if the network's width and depth are not decreased, classical deep network training can also benefit from encouraging flat representation manifolds. Less effective width can be achieved by setting some of the considered layer's weights to 0, which is exactly what the $L^1$-regularizer encourages. Less effective depth can be achieved by setting some layers to be the identity function. In a ReLU-network, a layer can mimic the identity function if the identity matrix is chosen as weight. The sparsity promoting $L^1$-regularizer could also encourage this behavior. Combining flatness promoting regularizers with the $L^1$ regularizer could therefore result in networks that effectively are less deep and wide, yielding better generalization for deep learning.

### 4.1.2 Special Case of flat or almost flat Manifolds

Theorem L4 (which is Theorem 4 in (Labate & Shi, 2024)) requires Assumption A 5a which prohibits the application of Theorem 3 to flat manifolds. To consider the case of flat manifolds, we replace Theorem L4 with a similar statement, but for flat manifolds. We need to make sure Assumption A 5 is still fulfilled, even if Assumptions A 5a and A 5b are not. However, as we will see in Corollary 1, for a flat manifold, we can choose $d_e = d$ which leads to a tighter upper bound than in the non-flat case (refer to Remark 7) .

Next, we study how far the manifold $\mathcal{M}$ can deviate from being flat while still having the tighter upper bound with $d_e = d$. This is considered in Corollary 2.

The proof of the following corollary is presented in the Appendix A.3.1.

**Corollary 1.** *If $\mathcal{M} \subset [0, \frac{1}{2\sqrt{D}}]^D$ is a flat manifold as described in Theorem 1, then there exists a matrix $A \in \mathbb{R}^{d \times D}$ and vector $c \in \mathbb{R}^d$ such that the projection $\Psi(x) = Ax + c$ fulfills*

$$||\Psi(x) - \Psi(y)||^2 = ||Ax - Ay||^2 = ||x - y||^2$$
$$and$$
$$\Psi(x) \in [0, 1]^d$$

*for all $x, y \in \mathcal{M}$.*

**Remark 7.** As by Corollary 1, if the considered manifold is flat, we can use $d$ as effective dimension, so $d_e = d$ while still fulfilling Assumption A 5. Let, for example, the intrinsic dimension be $d = 2$ and the ambient dimension be $D = 100$ of a flat manifold and Assumptions A 1-A 6 be fulfilled. Then, according to Theorem 3 using Corollary 1,

$$||\hat{f} - f_0||^2_{L^2(\mathcal{M}, \mu)} \leq 705.$$

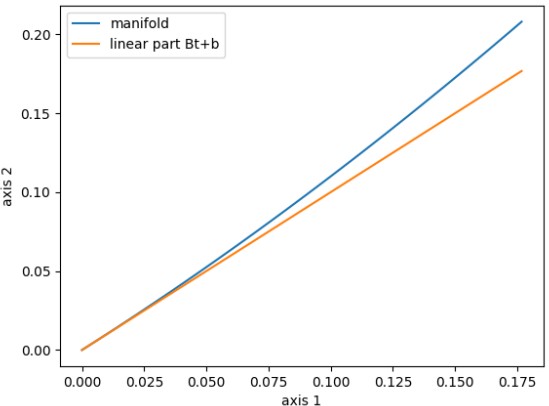

Figure 2: The manifold from Example 3 as well as its linear part.

For a non-flat manifold, we can use $d_e$ described in Assumption A 5b, leading to bounds that are extremely large so they become meaningless as described in Remark 4.

The following Corollary relaxes the flatness condition for achieving a significantly reduced upper bound. Again, the proof can be found in the Appendix A.3.2.

**Corollary 2.** *Let $\mathcal{M} \subset [0, \frac{1}{2\sqrt{D}}]^D$ be a compact $d$-dimensional Riemannian manifold which can be described by the parametrization $f \colon 0 \in \mathcal{M} = \{f(t) : t \in U \subset [0,1]^d, 0 \in U\}$ with $U$ convex. Fix $\epsilon \in \left(0, \frac{1}{3}\right]$.*
*For an orthonormal matrix $B \in \mathbb{R}^{D \times d}$ and a vector $b \in \mathbb{R}^D$, $f(t)$ can be partitioned in the following way:*

$$f(t) = b + Bt + r(t) \ , \tag{3}$$

*where $r$ describes the deviation of the manifold from the affine linear subspace given by $B$ and $b$.*
*If for all $x = f(t_x) \in \mathcal{M}$ and $y = f(t_y) \in \mathcal{M}$ the following holds:*

$$2||r(t_x) - r(t_y)||^2 \leq ||x - y||^2 \epsilon$$

*then there exists a vector $c \in \mathbb{R}^d$ so $\Psi(x) = B^T x + c$ fulfills Assumption A 5.*
*If further, Assumptions A 1-A 4 and A 6 are fulfilled, then Theorems 3 and 4 hold with $d_e = d$.*

**Example 3.** We know from Theorem 4, that there is a non-sharp upper bound which decreases with increasing flatness of the manifold. If the input manifold is close to being flat, Corollary 2 shows that we get an even lower bound as described in Remark 7. One example which satisfies the condition is the manifold

$$\mathcal{M} = \left\{ \left( \frac{t}{\sqrt{2}}, \frac{1}{\sqrt{2}}t + \frac{1}{2}t^2 \right)^T : t \in [0, 0.25] \right\},$$

which is a 1-dimensional manifold embedded in 2-dimensional space. The manifold is almost linear, but has a small deviation in the form of a quadratic term. Here,

$$B = \left( \frac{1}{\sqrt{2}}, \frac{1}{\sqrt{2}} \right)^T,$$
$$b = (0,0)^T,$$
$$r(t) = \left( 0, \frac{1}{2}t^2 \right)^T.$$

This manifold is displayed in Figure 2. With the projection $\Psi(x) = B^T x + 0$ for $x \in \mathcal{M}$, we get $\Psi(x) \in$

$[0, \frac{1}{4} + \frac{1}{32\sqrt{2}}] \subset [0, 1]$. We choose $\epsilon = \frac{1}{3} \in (0, \frac{1}{3}]$ and need to show $2||r(t_x) - r(t_y)||^2 \le \epsilon ||x - y||^2$ for all $t_x, t_y \in [0, 0.25]$ and $x = Bt_x + b + r(t_x)$, $y = Bt_y + b + r(t_y)$.

For $t_x = t_y$, we get $2||r(t_x) - r(t_y)||^2 = 0 = \epsilon ||x - y||^2$.

Now we consider the case $t_x \ne t_y$. Using the Mean Value Theorem, we get that there is a $\tilde{t} \in [0, 0.25]$ with the property

$$\frac{t_x^2 - t_y^2}{t_x - t_y} = 2\tilde{t}$$
$$\Leftrightarrow t_x^2 - t_y^2 = 2\tilde{t}(t_x - t_y)$$
$$\Rightarrow (t_x^2 - t_y^2)^2 = 4\tilde{t}^2(t_x - t_y)^2$$
$$\Rightarrow (t_x^2 - t_y^2)^2 \le \frac{4}{16}(t_x - t_y)^2 = \frac{1}{4}(t_x - t_y)^2. \tag{4}$$

Using this, we get

$$2||r(t_x) - r(t_y)||^2 = 2\frac{1}{4}(t_x^2 - t_y^2)^2$$
$$\overset{(4)}{\le} \frac{1}{8}(t_x - t_y)^2.$$

On the other hand,

$$\sqrt{\epsilon}||x - y|| = \sqrt{\epsilon}\left\|\frac{1}{\sqrt{2}}(t_x - t_y, t_x - t_y)^T + \left(0, \frac{1}{2}(t_x^2 - t_y^2)\right)^T\right\|$$
$$\ge \sqrt{\epsilon}\left(\left\|\frac{1}{\sqrt{2}}(t_x - t_y, t_x - t_y)^T\right\| - \left\|\left(0, \frac{1}{2}(t_x^2 - t_y^2)\right)^T\right\|\right)$$
$$= \sqrt{\epsilon}\left(|t_x - t_y| - \frac{1}{2}\sqrt{(t_x^2 - t_y^2)^2}\right).$$

Using that

$$\frac{1}{2}\sqrt{(t_x^2 - t_y^2)^2} \overset{(4)}{\le} \frac{1}{2}\sqrt{\frac{1}{4}(t_x - t_y)^2} = \frac{1}{4}|t_x - t_y|$$

yields

$$\sqrt{\epsilon}||x - y|| \ge \sqrt{\epsilon}\left(|t_x - t_y| - \frac{1}{4}|t_x - t_y|\right)$$
$$= \sqrt{\epsilon}\frac{3}{4}|t_x - t_y|.$$

Then,

$$\epsilon ||x - y||^2 \ge \frac{3}{16}(t_x - t_y)^2$$
$$\ge \frac{1}{8}(t_x - t_y)^2$$
$$\ge 2||r(t_x) - r(t_y)||^2.$$

The condition for Corollary 2 is fulfilled, leading to a smaller upper bound on the generalization gap for this manifold with $d_e = d$.

**Remark 8.** Note that the theorems and corollaries presented in this section have strict assumptions which are not fulfilled in many practical applications of NNs. Examples are Assumption A 5 about the structure of the considered NN, Assumption A 4 about the projection before the ReLU network, or the assumption from Corollaries 1 and 2 that $\mathcal{M} \subset [0, \frac{1}{2\sqrt{D}}]^D$ which is only a small interval for large $D$.

To check whether flat manifolds are still beneficial if these assumptions are not fulfilled, we conducted empirical experiments in the following Section 4.2.

### 4.2 Numerical Experiment

To show that flat manifolds can also be beneficial outside of the strict assumptions of Theorem 3, we conduct numerical experiments. First, we sample points on two manifolds of different curviness. Then we construct a ground truth function and apply it to the samples to receive our training and test data. Next, we create two identical neural networks and train them on the data from the two manifolds respectively and compare their performances.

To sample the points on the manifolds, we use the following steps:

1. Create $n$ random vectors in $\mathbb{R}^d$ with uniformly distributed entries between 0 and $2\pi$. This gives a matrix $R$ of size $(n \times d)$. These are the intrinsic coordinates of the manifold.

2. For each intrinsic dimension $i \in \{1, ..., d\}$, create a sine wave evaluated at the points $R[:, i]$ which means the $i$-th column. The frequency of the sine wave is $i \cdot c$ where $c > 0$ is a scalar which determines the curvature of the generated manifold. A small $c$ results in a manifold with higher reach (i.e. has less 'sharp' curves). The resulting vectors for all dimensions are added up. This vector is attached to $R$ as the column $d+1$, creating the matrix $\hat{M} \in \mathbb{R}^{n \times (d+1)}$, which acts as samples drawn from the manifold. The resulting manifold has $d$-dimensions and is embedded in $d+1$ dimensions with sine waves of different frequencies in each coordinate axis. The next step is to embed this manifold into the ambient space of dimension $D$.

3. Similar to what was done by Kienitz et al. (2022), we project the data to the ambient space using three matrices. The ambient coordinate matrix $\hat{M} \in \mathbb{R}^{n \times (d+1)}$, a matrix consisting of zeros $E = 0^{n \times (D-(d+1))}$ and a random orthogonal matrix $O \in \mathbb{R}^{n \times D}$. The final samples of the low dimensional manifold embedded into high dimensional space are computed as

$$X = \left[ \hat{M} \mid E \right] O^T \; \in \mathbb{R}^{n \times D},$$

where the first matrix in the product is a concatenation of $\hat{M}$ and $E$. The concatenation is the embedding into the $D$-dimensional space while the orthogonal matrix is used to rotate the resulting manifold.

Manifolds generated according to this procedure for $d = 2$, $D = 3$ and different values for $c$ are presented in Figure 3. For the numerical experiment, we created two manifolds in this way with $d = 10$, $D = 100$ and normalized the resulting coordinates to $[0, 1]^D$. The samples drawn from the first (less curvy) manifold, $X_1$ used $c = 0.1$ while the samples drawn from the second (more curvy) manifold, $X_2$ used $c = 3$. Then, we created target values by the following formula:

$$Y = \sin(X)W + b + z \in \mathbb{R} \; ,$$

where the entries in $W \in \mathbb{R}^{D \times 1}$ and $b \in \mathbb{R}$ are randomly drawn from a uniform distribution in the interval $[0, 1]$. The value $z$ is random gaussian noise distributed according to $\mathcal{N}(0, \sigma^2)$ with $\sigma^2 = 0.01$. The sine-function is applied elementwise to each element of the matrix $X$.

### 4.2.1 Experiment 1 with 10 neurons in the hidden layer

We created two identical neural networks with one hidden layer of size 10, which is the same as the intrinsic dimension $d$, and 1 neuron in the output layer and leaky ReLU activation. We trained one network on data from the curvy manifold $X_1$ and one on data from the less curvy manifold $X_2$. We used Mean Squared Error (MSE) as loss function, the Adam optimizer and 500 epochs with batch size 32. The whole process from the generation of the manifolds to the training and evaluation on a test set was repeated 10 times to alleviate the effect of random initializations.

The following results are summarized in Table 1. The test error of the network trained on $X_2$ was on average 0.113 while the error of the network trained on $X_1$ was on average 0.103. To check whether the networks'

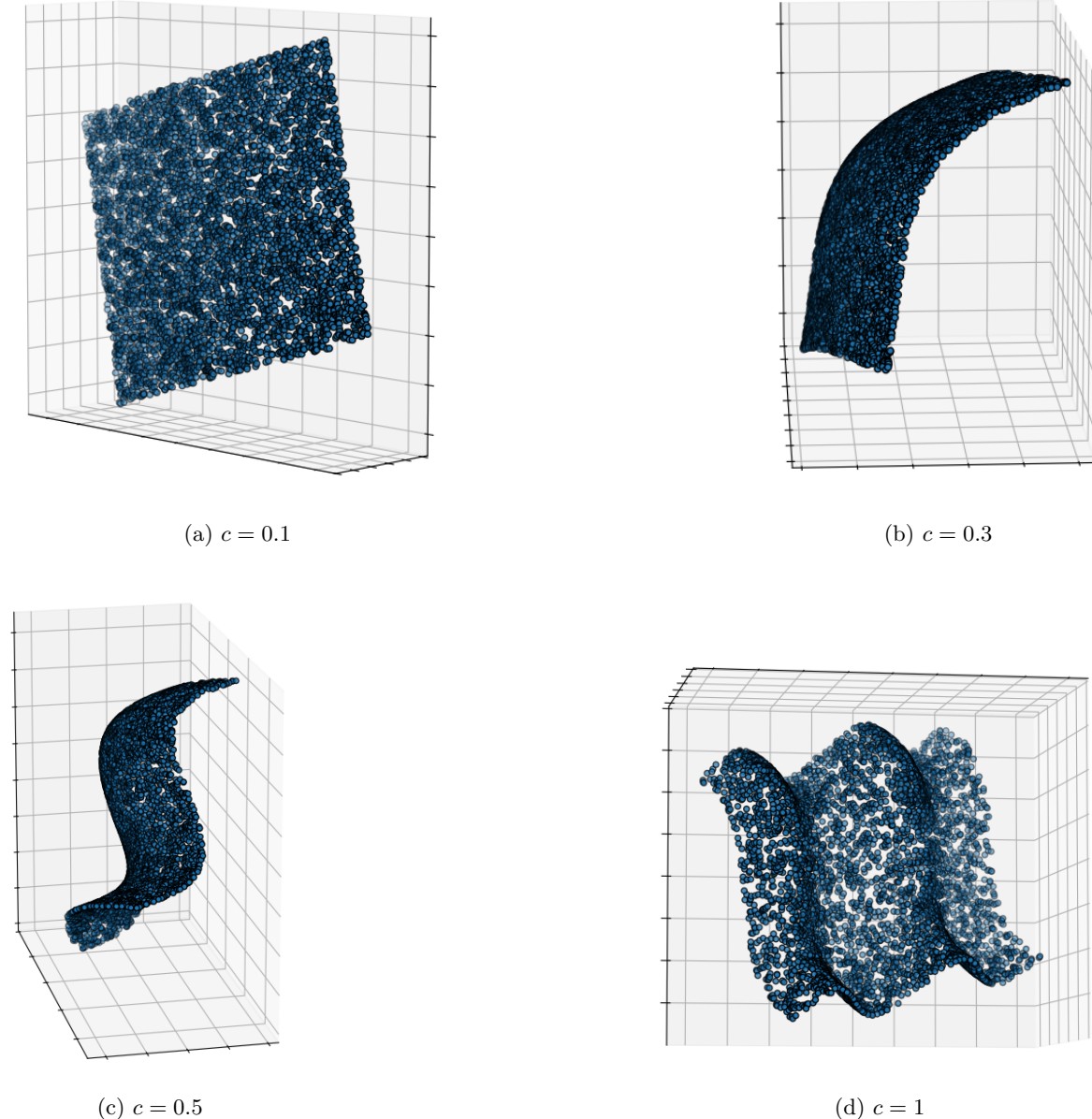

(a) $c = 0.1$

(b) $c = 0.3$

(c) $c = 0.5$

(d) $c = 1$

Figure 3: Manifolds with $d = 2$ and $D = 3$ and different curvatures.

Table 1: The test errors of the NN with 10 neurons in the hidden layer trained on flat and curvy input data. The test error of the base estimator, which simply predicts the average of the training output data, is also displayed. The p-value for the t-test with 0-hypothesis "The following are the same: The expected value of the test error of the NN trained on curvy manifold data and the expected value of the test error of the NN trained on flat manifold data." is 0.0002<0.05. The 0-hypothesis is rejected with 95% confidence level, showing that the difference in test error is statistically significant.

| manifold | mean test error of small NN | mean test error of base estimator |
|---|---|---|
| $X_1$ (less curvy) | 0.103 | 0.7 |
| $X_2$ (more curvy) | 0.113 | 0.7 |

Table 2: The test errors of the NN with 100 neurons in the hidden layer trained on flat and curvy input data with and without $L1$-regularization. The test error of the base estimator, which simply predicts the average of the training output data, is also displayed. The p-value for the t-test with 0-hypothesis "The following are the same: The expected value of the test error of the regularized NN trained on curvy manifold data and the expected value of the test error of the regularized NN trained on flat manifold data." is 0.02<0.05. The 0-hypothesis is rejected with 95% confidence level, showing that the difference in test error is statistically significant. The same test applied to the test errors of the NNs without regularization has a p-value of 0.25>0.05, meaning the difference is not statistically significant.

| manifold | mean test error of larger NN with regularization | mean test error of larger NN without regularization | mean test error of base estimator |
|---|---|---|---|
| $X_1$ (less curvy) | 0.137 | 0.116 | 0.6 |
| $X_2$ (more curvy) | 0.150 | 0.126 | 0.6 |

training can be considered successful on their own, we compared these results to a base estimator which predicted the average of $Y$ in the training set for all inputs. This estimator had an error of approximately 0.7 for both the flat manifold data ($X_1$) and the curvy manifold data ($X_2$). This is much higher than the networks' test error, meaning the training of both neural networks was overall successful. We performed a t-test with the 0-hypothesis "The following are the same: The expected value of the test error of the NN trained on curvy manifold data and the expected value of the test error of the NN trained on flat manifold data." The p-value was $0.0002 \leq 0.05$, indicating that the differences in test errors are not due to mere chance, but statistically significant.

This shows that in practice, flat input manifolds can be beneficial for training and generalization even in cases where the assumptions of Theorem 1 do not hold and the upper bound is not tight.

### 4.2.2 Experiment 2 with 100 neurons in the hidden layer

Usually in deep learning, wider networks are used than the one from Experiment 1. The transformer encoder model (Vaswani et al., 2017), for example, consists of sequential encoder blocks where the width of the input and output layers is the same. So, no dimensionality reduction is done. To check how such models are effected by more or less curvy manifolds, we repeated the steps described in Experiment 1, but with 5 repetitions and 100 neurons in the hidden layer of the used NNs. This is the same as the ambient dimension $D$. The results can be found in Table 2. The average error on the test set for $X_1$ was smaller than the average error for $X_2$, in accordance with the results from Experiment 1. However, when applying the t-test, the p-value was 0.25>0.05. The difference in test error was not statistically significant and as such we can not confidently say that the manifold's curviness has an impact on the generalization ability of the network. Inspired by Remark 6, we added an $L1$-regularizer to the weights of the model. The results for this can also be seen in Table 2. Here, the test error was again smaller for the network trained on $X_1$ than for the network trained on $X_2$. However, this time the t-test had a p-value of 0.02<0.05. Thus, the difference in test error was statistically significant. With 95% confidence we can say that a less curvy manifold leads to a better generalization performance. The number of weights less than $10^{-7}$ was increased by 15% for the network trained on $X_1$ compared to the network trained on $X_2$ which supports our hypothesis stated in Remark 6. Flat manifold datasets can be modeled by networks which have effectively less width (by setting weights to 0 during training), resulting in better generalization compared to more curvy manifold data.

## 5 Discussion

The theoretical framework established in our study indicates that flat representation manifolds, particularly in the second-to-last layer of classification networks, ensure perfect classification. This insight provides a mathematical foundation for the importance of flatness in classification tasks.

In the context of regression tasks, our result (Theorem 4) emphasizes that increasing flatness of input manifolds correlates with a reduction in the upper bound on the generalization gap. This relationship

suggests that when the underlying data manifold is less complex (characterized by larger reach) neural networks can better approximate target functions with fewer parameters. Consequently, this leads to enhanced performance and generalization capabilities. Further, when the manifold is close to being flat, the upper bound on the generalization gap reaches a significantly smaller value (Corollary 2 and Remark 7). The empirical evidence obtained from our numerical experiments supports the theoretical claims, revealing that neural networks trained on flat manifolds outperform those trained on more curvy counterparts, indicating the applicability of the claim outside the strict assumptions by Theorem 4. Combining flat manifolds with the $L^1$-regularizer extends the advantage of flat input manifolds to wide networks, hinting at the applicability of flatness promoting regularizers for deep network training.

## 6  Conclusion

Our study clearly demonstrates theoretical benefits of flat representation manifolds for both classification and regression tasks. Further, the benefits for regression tasks were confirmed empirically in settings which are closer to real-life applications of deep NNs than the theoretical results.

While our study demonstrates the benefits of flat manifolds, it is essential to acknowledge limitations. The presented theorems operate under specific assumptions which deviate from the usual applications, e.g. noiseless case in classification. As acknowledged in Remark 5, the bound in Theorem 4 is not tight and can become vacuous. Labate & Shi (2024) also acknowledge this but hope that a tighter upper bound can be found. One major point of interest for this is proving that Assumption A 5 can be fulfilled with smaller effective dimension $d_e$ in the general case (not only the flat or almost-flat case). This will be grounds for future research.

Further, the theorems do not yield practical flatness measures. In the case of regression, Theorem 4 considers the reach as flatness measure. However, since the reach is a maximum, it is hard to estimate from a limited amount of samples.

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

# A  Appendix

## A.1  Proof of Theorem 1

*Proof.* $(b) \Rightarrow (a)$
First we show that if the SFF vanishes, there exists $B \in \mathbb{R}^{D \times d}$ and $b \in \mathbb{R}^D$ fulfilling the claim.
If the SFF vanishes,

$$\frac{d}{dt}\Pi(\gamma(t))|_{t=0} = 0$$

for all $\gamma$ as in Definition 1 and all $p \in \mathcal{M}$. The projection $\Pi(p)$ onto the tangent space $T_p\mathcal{M}$ therefore does not depend on $p$, resulting in the same tangential space at all points $p \in \mathcal{M}$. Let $B \in \mathbb{R}^{D \times d}$ be a basis of the tangential space.
For a Riemannian manifold embedded in euclidean space, a vanishing SFF is equivalent to all geodesics being straight lines (Lee, 2018) (Proposition 8.12). The goedesics are the shortest paths between two points on the manifold. Therefore, and because $\mathcal{M}$ is simply connected, for all $y, \hat{y} \in \mathcal{M}$ we know that

$$\gamma(t) := \lambda\hat{y} + (1 - \lambda)y \in \mathcal{M}$$

is a path on the manifold, parametrized by $\lambda \in [0, 1]$. As such, $\dot{\gamma}(0) = \hat{y} - y$ is an element of the tangential space $T_y\mathcal{M}$ with basis $B$ and the following holds:

$$\forall \hat{y} \in \mathcal{M} \; \exists x \in \mathbb{R}^d : \hat{y} = \underbrace{Bx}_{=\hat{y}-y} + y = Bx + b.$$

We still need to show that $X := \{x : Bx + b \in \mathcal{M}\}$ is a convex set. Let $x_1, x_2 \in X$ and $y_1 = Bx_1 + b \in \mathcal{M}$, $y_2 = Bx_2 + b \in \mathcal{M}$. Let $\tilde{x} = \lambda x_1 + (1 - \lambda)x_2$ for some $\lambda \in [0, 1]$. From the convexity of $\mathcal{M}$ follows

$$B\tilde{x} + b = B(\lambda x_1 + (1 - \lambda)x_2) + b = \lambda B x_1 + (1 - \lambda)Bx_2 + b = \lambda y_1 + (1 - \lambda)y_2 \in \mathcal{M}.$$

Therefore, $\tilde{x} \in X$, proving the convexity of $X$.

$(a) \Rightarrow (b)$

Next we show that if $\mathcal{M} = \{Bx + b : x \in X\}$ for a convex set $X$, the SFF vanishes.
The tangential space is the same for every $p \in \mathcal{M}$ and given by the matrix $B$, so

$$T_p\mathcal{M} = \{Bx : x \in \mathbb{R}^d\}.$$

The matrix $\Pi(p) = B(B^TB)^{-1}B^T$ satisfies the two conditions that need to be fulfilled in order for $\Pi(p)$ to be the projection that defines the SFF (refer to Robbin & Salamon (2018)):

1. $\Pi(p)v = v \Leftrightarrow v \in T_p\mathcal{M}$
   If $\Pi(p)v = v$, then
   $$\Pi(p)v = B\underbrace{(B^TB)^{-1}B^Tv}_{:=w\in\mathbb{R}^d} = Bw,$$
   so $v$ is an element of the tangential space $T_p\mathcal{M}$.
   If on the other hand $v \in T_p\mathcal{M}$, then there exists a $w \in \mathbb{R}^d$ so $v = Bw$. Then
   $$\Pi(p)v = B(B^TB)^{-1}B^T\underbrace{Bw}_{=v} = Bw = v.$$

2. $\Pi(p) = \Pi(p)^2 = \Pi(p)^T$. This can be checked by inserting the definition of $\Pi(p)$ above.

Then, $d\Pi(p) = 0 \; \forall p$ since $\Pi(p)$ does not depend on $p$ and therefore $II(p) = 0$. $\qquad\qquad \square$

## A.2   Proof of Theorem 4

There are two proofs for this theorem. Both show that the bound in (1) can be upper bounded by a function which decreases monotonically with decreasing $d_e$ for large enough $n$.
The first proof is simpler, but the used upper bound is not tight to (1).
The second proof considers the upper bound of equation (1) that is the tightest possible differentiable function. By differentiation, we can see that the upper bound decreases with decreasing $d_e$. This proof is more elaborate, but also considers a tighter upper bound.

**Proof 1**

*Proof.* From the proof of the original theorem in Labate & Shi (2024) we know that the following upper bound holds with the specified probability:

$$||\hat{f} - f_0||^2_{L^2(\mathcal{M}, \mu)} \le (c_1 + c_2 d_e^\beta)n^{-2\beta/(2\beta+d_e)} \tag{5}$$

$$+ \frac{c_3}{n} W(d_e, n) \left( \ln(c_4 f(d_e, n)n^{\beta/(2\beta+d_e)}) + 1 \right)^2 \tag{6}$$

$$+ \frac{c_2}{n} W(d_e, n)d_e^\beta \ln\left( 4f(d_e, n)n^{\beta/(2\beta+d_e)} \right) \tag{7}$$

$$+ c_5 n^{-\beta/(2\beta+d_e)} \tag{8}$$

with

$$W(d_e, n) = (d_e + 1)N(d_e, n) + (L(d_e) - 1)N(d_e, n)^2 + L(d_e)N(d_e, n) + 1$$
$$f(d_e, n) = (L(d_e) + 1)(B(d_e, n) + 2)B(d_e, n)^{L(d_e)+1}N(d_e, n)^{L(d_e)+1}$$
$$N(d_e, n) = 2^{d_e/\beta+1}(6d_e + 47)n^{d_e/(4\beta+2d_e)}$$
$$L(d_e) = (28d_e^2 - 15)d_e^{d_e/4}(c_7d_e^\beta + 6)^{d_e/(2\beta)}$$
$$B(d_e, n) = \left\lceil c_6 2^{d_e/\beta} n^{1/(2\beta+d_e)}(\ln(n^{d_e/(4\beta+2d_e)} + 2))^{1/d_e} \right\rceil \left\lceil d_e^{1/2}(c_7d_e^\beta + 6)^{1/\beta} \right\rceil.$$

It is sufficient to show that terms (5), (6), (7) and (8) grow monotonically with increasing $d_e$ or can be upper bounded by a function that does so. For terms (5) and (8) this is clear. We consider terms (6) and (7) more thoroughly.

Since $N(d_e, n)$ and $L(d_e)$ increase monotonically with $d_e$ and $L(d) > 1$ for $d_e \geq 1$, $W(d_e, n)$ also grows monotonically with $d_e$. The only critical part in term (6) is $f(d_e, n)n^{\beta/(2\beta+d_e)}$.

First, we find an upper bound for $f(d_e, n)$:

$$f(d_e, n) \leq 4L(d_e)B(d_e, n)^{L(d_e)+2}N(d_e, n)^{L(d_e)+1}. \tag{9}$$

Here we used that $L(d_e) \geq 1$ and $B(d_e, n) \geq 2$ for large enough $n$. Now, we consider the functions $L$ and $B$ separately. $L(d_e)$ increases monotonically with $d_e$ and is upper bounded by,

$$L(d_e) \leq 28d_e^2 \, l(d_e)$$

where $l(d_e) = d_e^{d_e/4}(c_7d_e^\beta + 6)^{d_e/(2\beta)}$. Let $b(d_e) := c_6 2^{d_e/\beta}(d_e^{1/2}(c_7d_e^\beta + 6)^{1/\beta})$.
Then,

$$B(d_e, n) = n^{1/(2\beta+d_e)}(\ln(n^{d_e/(4\beta+2d_e)} + 2))^{1/d_e}b(d_e) \leq n^{2/(2\beta+d_e)}b(d_e)$$

for large enough $n$ for which $n^{1/(2\beta+d_e)} \geq (\ln(n^{d_e/(4\beta+2d_e)} + 2))^{1/d_e}$ holds. Here, $b(d_e)$ is monotonically increasing with $d_e$. Therefore,

$$B(d_e, n)^{L(d_e)+2} \leq n^{(56d_e^2 \, l(d_e)+4)/(2\beta+de)}b(d_e)^{L+2}.$$

Using equation (9), we get:

$$f(d_e, n) \leq \underbrace{4L(d_e)2^{(L+1)(d_e/\beta+1)}(6d_e + 47)^{L+1}b(d_e)^{L+2}}_{=:g(d_e,n)} n^{(28d_e^3 \, l(d_e)+112d_e^2 \, l(d_e)+2d_e+8)/(4\beta+2d_e)}$$

where $g$ is monotonically increasing with $d_e$. Then,

$$f(d_e, n)n^{\beta/(2\beta+d_e)} \leq g(d_e, n)n^{h(d_e)}$$
$$h(d_e) := \frac{28d_e^3 \, l(d_e) + 112d_e^2 \, l(d_e) + 2d_e + 8 + 2\beta}{4\beta + 2d_e}.$$

The function $h$ is increasing monotonically with $d_e$, making the whole expression increase monotonically. Therefore, there is an upper bound on term (6) which decreases with decreasing $d_e$.

The same can be used to show that (7) is upper bounded by a function which increases monotonically with $d_e$.

Theorem 3 and therefore this derivation hold with probability at least $1 - 2\exp(-n^{\frac{d_e}{2\beta+d_e}})$. As this approaches 1 for large $n$, we arrive at the conclusion of the proof. $\square$

### Proof 2

*Proof.* This more elaborate proof consists of three steps:

1. Find an upper bound for the bound from Theorem 3 using a differentiable function. Mainly, this means getting rid of ceiling operations by using $\lceil k \rceil \leq k + 1$ for $k \in \mathbb{R}$.

2. Compute the derivative of the bound with respect to $d_e$ and show that it is positive for large sample sizes $n$. With increasing flatness of the manifold, the reach $\tau$ increases, decreasing $d_e$. Due to the positive derivative, this results in a decreasing bound.

3. Theorem 3 and therefore this derivation hold with probability at least $1 - 2\exp(-n^{\frac{d_e}{2\beta+d_e}})$. As this approaches 1 for large $n$, we arrive at the conclusion of the proof.

From the proof of the original theorem in Labate & Shi (2024) we know that the following upper bound holds with the specified probability:

$$||\hat{f} - f_0||_{L^2(\mathcal{M},\mu)}^2 \leq (c_1 + c_2\tilde{d}_e^{\,\beta})n^{-2\beta/(2\beta+\tilde{d}_e)} \tag{10}$$

$$+ \frac{c_3}{n}W(\tilde{d}_e, n)\left(\ln(c_4 f(\tilde{d}_e, n)n^{\beta/(2\beta+\tilde{d}_e)}) + 1\right)^2 \tag{11}$$

$$+ \frac{c_2}{n}W(\tilde{d}_e, n)\tilde{d}_e^{\,\beta}\ln\left(4f(\tilde{d}_e, n)n^{\beta/(2\beta+\tilde{d}_e)}\right) \tag{12}$$

$$+ c_5 n^{-\beta/(2\beta+\tilde{d}_e)} \tag{13}$$

$$=: b(\lceil d_e \rceil, \lceil B_1(\lceil d_e \rceil, n) \rceil \cdot \lceil B_2(\lceil d_e \rceil, n) \rceil) \tag{14}$$

with

$$\tilde{d}_e = \lceil d_e \rceil$$
$$W(d_e, n) = (d_e + 1)N(d_e, n) + (L(d_e) - 1)N(d_e, n)^2 + L(d_e)N(d_e, n) + 1$$
$$f(d_e, n) = (L(d_e) + 1)(B(d_e, n) + 2)B(d_e, n)^{L(d_e)+1}N(d_e, n)^{L(d_e)+1}$$
$$N(d_e, n) = 2^{1+d_e/\beta}(6d_e + 47)n^{d_e/(4\beta+2d_e)}$$
$$L(d_e) = (28d_e^2 - 15)d_e^{d_e/4}(c_7 d_e^\beta + 6)^{d_e/(2\beta)}$$
$$B(d_e, n) = \Big\lceil \underbrace{c_6 2^{d_e/\beta} n^{1/(2\beta+d_e)}(\ln(n^{d_e/(4\beta+2d_e)} + 2))^{1/d_e}}_{B_1}\Big\rceil \Big\lceil \underbrace{d_e^{1/2}(c_7 d_e^\beta + 6)^{1/\beta}}_{B_2}\Big\rceil.$$

In the term above and the rest of the proof, the constants $c_i$ are positive and independent of $d_e$ and $n$. They can depend on the diameter of the manifold, and properties of $f_0$ like $\beta$ and $R$. Similar, constants $b_i$ are positive and do not depend on $n$.

To prove the statement, we show that the derivative of the upper bound with respect to the effective dimension $d_e$ is positive for all $n \geq \hat{N}$. With increasing reach, i.e. increasing flatness of the manifold, $d_e$ decreases. So if the upper bound's derivative with respect to $d_e$ is positive, then an increasing reach leads to a decreased upper bound.

We want to show that $b$ can be bounded from above by a function that has a positive derivative with respect to $d_e$ (step 2). First, as described in step 1, we have to find this differentiable bounding function (note that claims (i) and (ii) are proven later on):

$$b(\lceil d_e \rceil, \lceil B_1(\lceil d_e \rceil) \rceil \cdot \lceil B_2(\lceil d_e \rceil) \rceil) \overset{(i)}{\leq} b(d_e + 1, (B_1(\lceil d_e \rceil) + 1) \cdot (B_2(\lceil d_e \rceil) + 1))$$

$$\overset{(ii)}{\leq} b(d_e + 1, (B_1(d_e - 1) + 1) \cdot (B_2(d_e + 1) + 1))$$

$$= b(h(d_e), \underbrace{h(B_1(l(d_e))) \cdot h(B_2(h(d_e)))}_{y(h(B_1), h(B_2))})$$

where $h(x) := x + 1$ and $l(x) := x - 1$. In the course of the proof we will show that $g_1(d_e, n) := \frac{\partial b(d_e, y)}{\partial d_e} \geq 0$, $g_2(d_e, n) := \frac{\partial b(d_e, y)}{\partial y} \geq 0$, $g_3(d_e, n) := \frac{\partial y(B_1, B_2)}{\partial B_1} \geq 0$ and $g_5(d_e, n) := \frac{\partial y(B_1, B_2)}{\partial B_2} \geq 0$, proving claim (i).

Further, to prove claim (ii), we show $g_4 := \frac{\partial B_1(d_e)}{\partial d_e} \leq 0$ and $g_6 := \frac{\partial B_2(d_e)}{\partial d_e} \geq 0$.
We want to show that the derivative of $b$ with respect to $d_e$ is positive. The total derivative of $b$ with respect to $d_e$ can be computed as:

$$\frac{d\ b(h(d_e), y(h(B_1(l(d_e))), h(B_2(h(d_e)))))}{d\ d_e} = \frac{\partial b(h(d_e), y)}{\partial h(d_e)} \underbrace{\frac{\partial h(d_e)}{\partial d_e}}_{=1}$$

$$+ \frac{\partial b(h(d_e), y)}{\partial y} \left( \frac{\partial y(h(B_1), h(B_2))}{\partial h(B_1)} \underbrace{\frac{\partial h(B_1)}{\partial B_1}}_{=1} \frac{\partial B_1(l(d_e))}{\partial l(d_e)} \underbrace{\frac{\partial l(d_e)}{\partial(d_e)}}_{=1} \right.$$

$$\left. + \frac{\partial y(h(B_1), h(B_2))}{\partial h(B_2)} \underbrace{\frac{\partial h(B_2)}{\partial B_2}}_{=1} \frac{\partial B_2(h(d_e))}{\partial h(d_e)} \underbrace{\frac{\partial h(d_e)}{\partial d_e}}_{=1} \right)$$

$$= \underbrace{\frac{\partial b(h(d_e), y)}{\partial h(d_e)}}_{g_1(d_e)} + \underbrace{\frac{\partial b(h(d_e), y)}{\partial y}}_{g_2(d_e)} \left( \underbrace{\frac{\partial y(h(B_1), h(B_2))}{\partial h(B_1)}}_{g_3(d_e)} \underbrace{\frac{\partial B_1(l(d_e))}{\partial l(d_e)}}_{g_4(d_e)} \right.$$

$$\left. + \underbrace{\frac{\partial y(h(B_1), h(B_2))}{\partial h(B_2)}}_{g_5(d_e)} \underbrace{\frac{\partial B_2(h(d_e))}{\partial h(d_e)}}_{g_6(d_e)} \right)$$

$$\overset{(a)}{\geq} g_1(d_e, n) + g_2(d_e, n) g_3(d_e, n) g_4(d_e, n)$$

$$\overset{(b)}{\geq} 0.$$

We have to prove $(a)$ and $(b)$.
Throughout the whole proof, we will use the following variables in the exponents:

$$b_5 := \frac{\beta}{2\beta + d_e}$$

$$b_{17} := \frac{d_e}{4\beta + 2d_e}$$

$$b_{30} := \frac{1}{2\beta + d_e}$$

$$2b_{17} - 1 = -2b_5 = b_{30} - \frac{2\beta + 1}{2\beta + d_e} \tag{15}$$

We omit the precise definitions of the other variables $b_i > 0$ since defining them would not contribute to the arguments of the proof.
First, we show claim $(b)$. We compute the asymptotic behavior of $g_1$, $g_2$, $g_3$ and $g_4$ and compare positive and negative contributions of the term $g_1(d_e, n) + g_2(d_e, n) g_3(d_e, n) g_4(d_e, n)$.

**Closed form and asymptotic behavior of $g_1$**

We consider the closed form

$$
\begin{aligned}
g_1(d_e, n) =\ & n^{-b_5}\left(b_1 \ln(n) + b_2 n^{-b_5}\ln(n) + b_4 n^{-b_5}\right) \\
& + \underbrace{\left[n^{b_{17}}(b_{13} + b_{14}\ln(n)) + n^{2b_{17}}(b_{15} + b_{16}\ln(n))\right]}_{\partial_{d_e} W(d_e, n)}\left[\frac{c_3}{n}(\ln(c_4 f(d_e, B_1, B_2)n^{b_5}) + 1)^2 + \right. \\
& \left. + \frac{c_2}{n}b_7 \ln(4 f(d_e, B_1, B_2)n^{b_5})\right] \\
& + \underbrace{(1 + b_{11}n^{b_{17}} + b_{12}n^{2b_{17}})}_{W(d_e, n)}\left(\frac{c_3}{n}2\left(\ln(c_4 f(d_e, B_1, B_2)n^{b_5}) + 1\right)\left(\frac{\partial_{d_e} f(d_e, B_1, B_2)}{f(d_e)} - b_9 \ln(n)\right)\right. \\
& \left. + \frac{c_2}{n}\left(b_{10}(\ln(4 f(d_e, B_1, B_2)n^{b_5})) + b_7\left(\frac{\partial_{d_e} f(d_e, B_1, B_2)}{f(d_e, B_1, B_2)} - b_9 \ln(n)\right)\right)\right).
\end{aligned}
\tag{16}
$$

Further,

$$
\begin{aligned}
\ln(cf(d_e, B_1, B_2)n^{b_5}) &= \ln(c) + \ln(f(d_e, B_1, B_2)) + b_5 \ln(n) \\
&= \ln(c) + b_5 \ln(n) + \ln(L(d_e) + 1) + \ln(B(d_e, n) + 2) + (L(d_e) + 1)(\ln(B(d_e, n)) + \ln(N(d_e, n))) \\
&= a + b_6 \ln(n) + b_{18}\ln(\ln(n^{b_{17}} + 2)) + \ln(B(d_e, n) + 2) \\
&\geq a + b_6 \ln(n) + b_{18}\ln(\ln(n^{b_{17}} + 2)) \geq 0 \text{ for large } n,
\end{aligned}
$$

Here, $a$ is a constant with respect to $n$ which can be negative. In terms of asymptotic behavior, $\ln(cf(d_e, B_1, B_2)n^{b_5}) \in \mathcal{O}(\ln(n) + \ln(\ln(n)))$. Note that we use the $\mathcal{O}$-notation here and in the rest of the proof to describe asymptotic behavior for the case $n \to \infty$.
Next, we compute a closed expression for:

$$
\frac{\partial_{d_e} f(d_e, B_1, B_2)}{f(d_e, B_1, B_2)} = b_{22} + b_{23}\ln(n) + b_{24}\ln(\ln(n^{b_{17}} + 2)).
$$

Note that the first term in the sum in equation (16) is positive for all $n \geq 1$. The second term grows with

$$
\mathcal{O}(n^{2b_{17}-1}\ln(n)^3).
$$

The third term in (16) could be negative and would be decreasing at most with

$$
\mathcal{O}(n^{2b_{17}-1}\ln(n)^2).
$$

Therefore, with regards to $n$,

$$
g_1(d_e, n) \in \mathcal{O}(n^{2b_{17}-1}\ln(n)^3 - n^{2b_{17}-1}\ln(n)^2).
\tag{17}
$$

**Closed form and asymptotic behavior of $g_2$**
We consider the closed form

$$
g_2(d_e, n) = \underbrace{\frac{\partial_y f(h(d_e), y)}{f(h(d_e), y)}}_{(a)} W(d_e, n)\left(\underbrace{2\frac{c_3}{n}\left(\ln(c_4 f(h(d_e), y)n^{b_5}) + 1\right) + \frac{c_2}{n}d_e^\beta}_{(b)}\right).
$$

The term $(b)$ is positive for all $n \geq 1$, the same holds for $W$ which grows with $n^{2b_{17}}$. Let's consider term $(a)$:

$$
\frac{\partial_y f(h(d_e), y)}{f(h(d_e), y)} = \frac{1}{y+2} + \frac{L+1}{y}.
$$

The term $y = B_1(d_e) \cdot B_2(d_e)$ is positive for all $n \geq 1$. $L$ is also positive, making the whole term positive. Using that $\frac{1}{B+2} + \frac{L+1}{B} \leq \frac{L+2}{B}$, we can deduce that

$$g_2(d_e, n) \in \mathcal{O}\left(\frac{\ln(n)}{\ln(n^{b_{17}} + 2)^{1/h} n^{(2\beta+1)/(2\beta+d_e)}}\right).$$

**Closed form and asymptotic behavior of $g_3$**

We consider the closed form

$$g_3(d_e, n) = \frac{\partial(h(B_1)h(B_2))}{\partial h(B_1)} = h(B_2) = b_{31},$$

which is positive.

**Closed form and asymptotic behavior of $g_4$**

We consider the closed form

$$\frac{\partial B_1(h)}{\partial h} = b_{25} n^{b_{30}} \left(\ln(n^{b_{17}} + 2)^{1/h}\right) \left(b_{26} - b_5^2 \ln(n) - b_{27}^2 \ln(\ln(n^{b_{17}} + 2)) + b_{28} \frac{\ln(n)}{\ln(n^{b_{17}} + 2)} \frac{n^{b_{17}}}{n^{b_{17}} + 2}\right),$$

which is negative for large $n$. In the computation of the derivative, $g_4$ is multiplied by $g_2$ and $g_3$. We want to show that this negative product grows slower than the positive term $g_1$, which would guarantee a positive value for the sum of both terms for large enough $n$.

**Combining all four terms**

The negative term of the product $g_2(d_e, n)g_3(d_e, n)g_4(d_e, n)$ grows with $\ln(n)^2 n^{-2b_5}$. At the same time, $g_1 \in \mathcal{O}(\ln(n)^3 n^{-2b_5} - \ln(n)^2 n^{-2b_5})$ (refer to (17) and (15)), making the positive contribution larger than the negative one for large $n$. This proves claim (b) for large enough $n$.

To prove claim $(a)$, we show that $g_2(d_e, n), g_5(d_e, n)$ and $g_6(d_e, n)$ are positive:

$$g_5(d_e, n) = h(B_1) \geq 0$$
$$g_6(d_e, n) = b_{29} \geq 0.$$

We showed prior that $g_2(d_e, n) \geq 0$. Together, these prove claim $(a)$, thereby concluding the proof. $\qquad\square$

## A.3 Proofs of Corollaries

### A.3.1 Proof of Corollary 1

*Proof.* Remember that $||.|| = ||.||_2$ is the euclidean norm either in matrix or vector form.
If $\mathcal{M}$ is an affine linear space, then there exists a matrix $B \in \mathbb{R}^{D \times d}$ and a vector $b \in \mathbb{R}^D$ so for all $x \in \mathcal{M}$ there exists a $t \in \mathbb{R}^d$ so $x = Bt + b$. We can assume that the matrix $B$ consists of the normalized basis vectors of the subspace, so $B^T B = I \in \mathbb{R}^{d \times d}$ is the identity matrix.
Take $A = B^T$ as the projection matrix, defining the projection as $\Psi(x) = B^T x + c$ for a suitable $c \in \mathbb{R}^d$ so $\Psi(x) \in [0, 1]^d \ \forall x \in \mathcal{M}$. We will later show that such a $c$ exists.
Then for all $x, y \in \mathcal{M}$:

$$||\Psi(x) - \Psi(y)||^2 = ||B^T \underbrace{x}_{=Bt_1+b} + c - B^T \underbrace{y}_{=Bt_2+b} - c||^2$$
$$= ||B^T B t_1 + B^T b - B^T B t_2 - B^T b||^2$$
$$= ||B^T B(t_1 - t_2)||^2 = ||t_1 - t_2||^2 = (t_1 - t_2)^T B^T B(t_1 - t_2)$$
$$= ||B(t_1 - t_2)||^2 = ||Bt_1 + b - Bt_2 - b||^2 = ||x - y||^2.$$

Let's check that $\Psi(x) \in [0,1]^{d_e}$: Since $B$ consists of normalized vectors and $A = B^T$,

$$\sum_{i=1}^{D} A_{j,i}^2 = 1 \quad \forall j = 1, ..., d.$$

The maximum of $\sum_{i=1}^{D} A_{j,i} x_i$ for any $x \in \left[0, \frac{1}{2\sqrt{D}}\right]^D$ is reached for $A_{j,i} = \frac{1}{\sqrt{D}}$ and $x_i = \frac{1}{2\sqrt{D}}$ $\forall i$, resulting in

$$\sum_{i=1}^{D} A_{j,i} x_i \leq \frac{1}{2}.$$

To show this, we have to solve the following optimization problem for some $j$:

$$\max_{x_i, A_{j,i}} \sum_{i=1}^{D} A_{j,i} x_i$$
$$\text{s.t.} - x_i \leq 0 \ \forall i$$
$$x_i - \frac{1}{2\sqrt{D}} \leq 0 \ \forall i$$
$$\sum_{i=1}^{D} A_{j,i}^2 - 1 = 0$$

The KKT-conditions (refer to Theorem 12.1 in (Nocedal & Wright, 2006)) state that the optimal $x_i$ and $A_{j,i}$ for all $i$ solves the following system of equations:

1. (Stationarity) For $\mu, \tilde{\mu} \in \mathbb{R}^D$ and $\lambda \in \mathbb{R}$, the gradient with respect to $x$ and $A_{j,.}$ of the function

$$\mathcal{L}(x, A_{j,.}, \mu, \tilde{\mu}, \lambda) = -\sum_i A_{j,i} x_i + \sum_i \mu_i(-x_i) + \sum_i \tilde{\mu}_i \left(x_i - \frac{1}{2\sqrt{D}}\right) + \lambda \left(\left(\sum_i A_{j,i}^2\right) - 1\right)$$

   has to be 0. This means

$$- A_{j,i} - \mu_i + \tilde{\mu}_i = 0 \ \forall i$$
$$- x_i + 2\lambda A_{j,i} = 0 \ \forall i.$$

2. (Primal Feasibility 1) The equality condition has to be met: $\sum_{i=1}^{D} A_{j,i}^2 = 1$

3. (Primal Feasibility 2) The inequality conditions have to be met:

$$- x_i \leq 0 \ \forall i$$
$$x_i - \frac{1}{2\sqrt{D}} \leq 0 \ \forall i$$

4. (Dual Feasibility) $\mu_i \geq 0$ and $\tilde{\mu}_i \geq 0$ $\forall i$

5. (Complementarity)

$$\mu_i(-x_i) = 0 \ \forall i$$
$$\tilde{\mu}_i \left(x_i - \frac{1}{2\sqrt{D}}\right) = 0 \forall i$$

When solving this system of equations, the possible candidates for the maximum are characterized by the following:

$$I \subseteq \{1, ..., D\}$$
$$K \subseteq \{1, ..., D\}$$
$$I \cup K = \{1, ..., D\}$$
$$x_i = 0 \ \forall i \in I$$
$$x_k = \frac{1}{2\sqrt{D}} \ \forall k \in K$$
$$A_{j,i} = 0 \ \forall i \in I$$
$$A_{j,k} = \frac{1}{\sqrt{|K|}} \ \forall k \in K,$$

where $|K|$ is the number of elements in the index set $K$.

The objective function values of these solutions are determined by $|I|$ and $|\mathrm{K}|$ and given by

$$\sum_i A_{j,i} x_i = \sum_{i \in I} 0 + \sum_{k \in K} \frac{1}{2\sqrt{|K|}\sqrt{D}} = \frac{1}{2} \frac{\sqrt{|K|}}{\sqrt{|D|}}.$$

The largest objective function value is reached for $|K| = D$ and therefore $|I| = 0$. The maximal value is

$$\sum_i A_{j,i} x_i = \frac{1}{2} \frac{\sqrt{|D|}}{\sqrt{|D|}} = \frac{1}{2}.$$

With a similar approach, one can show that the minimum is reached by $A_{i,j} = -\frac{1}{\sqrt{D}}$, resulting in

$$-\frac{1}{2} \le \sum_{i=1}^{D} A_{j,i} x_i.$$

Defining the full projection as $\Psi(x) = Ax + c = Ax + \frac{1}{2}[1, ..., 1]^T$ yields $\Psi(x) \in [0, 1]^{d_e}$. $\qquad\square$

### A.3.2 Proof of Corollary 2

*Proof.* Remember that $||.|| = ||.||_2$ is the euclidean norm either in matrix or vector form.

We have to show that Assumption A 5 holds for the projection with $A = B^T$ defined as $\Psi(x) = B^T x + c$. The assumption states the following two properties

C 1 $(1 - \epsilon)||x - y|| \le ||\Psi(x) - \Psi(y)|| = ||B^T x + c - B^T y - c|| = ||B^T x - B^T y|| \le (1 + \epsilon)||x - y||$ for all $x, y \in \mathcal{M}$.

C 2 $\Psi(x) \in [0, 1]^{d_e} \ \forall x \in \mathcal{M}$.

With the same argument as in the proof of Corollary 1, Consition C 2 is fulfilled for $B^T$ as projection matrix and a suitably chosen vector $c$.

Using $B^T$ as projection matrix yields for $x = f(t_x) \in \mathcal{M}$ and $y = f(t_y) \in \mathcal{M}$:

$$\begin{aligned}
||B^T x - B^T y||^2 &= ||B^T b - B^T b + B^T B t_x - B^T B t_y + B^T r(t_x) - B^T r(t_y)||^2 \\
&\le ||B^T B t_x - B^T B t_y||^2 + ||B^T||^2 \ ||r(t_x) - r(t_y)||^2 \\
&= ||B t_x + b + r(t_x) - B t_y - b - r(t_y) - r(t_x) + r(t_y)||^2 + ||B^T||^2 \ ||r(t_x) - r(t_y)||^2 \\
&\le ||x - y||^2 + ||r(t_x) - r(t_y)||^2 + ||B^T||^2 \ ||r(t_x) - r(t_y)||^2.
\end{aligned}$$

Note that in the second line we use $||Ax||_2 \leq ||A||_2||x||_2$ for a matrix $A$ and vector $x$. The third line follows from similar arguments as presented in the proof of Corollary 1.

Inserting

$$||B^T||^2 = ||B^T||_2^2 = ||B||_2^2 = \lambda_{\max}(B^T B) = \lambda_{\max}(I) = 1,$$

where $\lambda_{\max}$ is the operator returning the maximal eigenvalue, yields

$$||B^T x - B^T y||^2 \leq ||x - y||^2 + 2||r(t_x) - r(t_y)||^2$$
$$\leq ||x - y||^2(1 + \epsilon) .$$

Similarly, using $||x - y||^2 \geq ||x||^2 - ||y||^2$, we get

$$||B^T x - B^T y||^2 = ||B^T b - B^T b + B^T B t_x - B^T B t_y + B^T r(t_x) - B^T r(t_y)||^2$$
$$\geq ||B^T B t_x - B^T B t_y||^2 - ||B^T||^2||r(t_y) - r(t_x)||^2$$
$$\geq ||x - y||^2 - 2||r(t_y) - r(t_x)||^2$$
$$\geq ||x - y||^2(1 - \epsilon) . \tag{18}$$

To conclude the proof note that from (18) follows:

$$\underbrace{\sqrt{1 - \epsilon}}_{:=1-\tilde{\epsilon}\geq 1-\epsilon} \ ||x - y|| \leq ||B^T x - B^T y||$$

Therefore, there exists an $\tilde{\epsilon} \leq \epsilon \leq \frac{1}{3}$ for which Condition C 1 holds for all $x, y$ on the manifold. $\qquad \square$

