# OpenReview forum: "On the Generalization Superiority of Flat Representation Manifolds for Deep Learning Machines"
_TMLR — Rejected by TMLR_

### Review · Reviewer_hztr · 2025-12-21

**Summary Of Contributions:**

The results of this paper are mainly on discussing how the manifold structure of samples affects the bounds of estimates of generalization error of neural networks. In particular, it shows that the (upper) bound of Theorem 3 decreases as the effective dimension $d_e$ decreases, and this can be achieved by e.g. flattening the sample input manifold. Examples are given to discuss the conditions on flattening. Then experiments are given to illustrate the theorems.

**Audience:**

Yes

**Audience Explanation:**

The authors propose an interesting question.

**Broader Impact Concerns:**

N.A.

**Claims And Evidence:**

No

**Claims Explanation:**

While the topic of this paper is interesting, many things in the paper are not made clear. For example:

1. The network structure based on which the theory develops is unclear. I can only see that it is a “ReLU network” with some requirements on the layers, etc. It is not clear how the projection map is “followed by” the ReLU network, either.

2. The definition of “argmax” in Theorem 2 is unclear. For example, if the vector is $v = (1,1,1)$ what is “the index” that the argmax gives?
3. In Definition 2, I guess the x and x are used interchangeably.

4. In Assumption A4, I guess the $\mathcal{R}$ and R are used interchangeably.

5. The definition of $d_e$ in Assumption A5 is unclear. Moreover, since A is a $d\times D$ matrix and c a vector in $\mathbb{R}^d$, it seems that $d = d_e$ must hold. But then this contradicts Theorem L4 (A5 (b)).

6. The assumptions of Theorem L4 and thus Theorem 4 are not applicable for compact flat manifolds, while Theorem 2 assumes compact flat manifolds. I wonder how they are related to one another?

7. In Remark 5, $d_e$ is referred to as “projection dimension” instead of effective dimension.

8. In Corollary 1, an affine linear subspace is unbounded, thus cannot be contained in the cube $[0, 1/2\sqrt{D}]^D$.
9. In Corollary 2, $\mathcal{M}$ is required to be a compact manifold in $[0, 1/2\sqrt{D}]^D$. When D is large (which is the case when low-dimensional structure of data becomes interesting), this is a small set, and may not represent the real case.

10. The experiments in Section 4.2.1 use Leaky ReLU instead f ReLU, thus cannot back the theoretical results.

There are other unclear things. Because of such confusion, I am unable to check the proofs and other technical things.

Besides, the paper aims at showing the decrease of upper bound of the generalization error. Unless the generalization is of the same order as the upper bound, the generalization error may not change at all! Indeed, the paper proposes an interesting point of view, but it is not enough to explain what the title suggests, namely generalization would be better for flatter manifolds.

**Requested Changes:**

Please add sufficient details to the notations and arguments. It would be better if you could strengthen the results.

---

> ### Author Response · Authors · 2026-01-14
> **Answer to Review by Reviewer hztr**
>
> (I will try to split the answer into two parts due to the length restriction)
> Dear reviewer,
> Thank you for your thorough consideration of our manuscript and the detailed review. I would like to provide further clarification by providing the following responses:
>
> •	“The network structure based on which the theory develops is unclear. I can only see that it is a “ReLU network” with some requirements on the layers, etc. It is not clear how the projection map is “followed by” the ReLU network, either.”
>
> Thank you for pointing out this unprecise formulation.
> To address this, I specified in Assumption A6 “A ReLU network is a fully connected feed forward NN where the activation function is the well-known rectified linear unit function.” Further, I added “maximal norm of the weights B ∈ R.” I think the target group of readers should be familiar with the notion of ReLU networks. Since I clarify that these assumptions are inherited from Labate et al.’s work, readers can also refer to this article for more clarification.
> Further, I added “This means that the output of the projection is the input for the ReLU NN further detailed in Assumption A 6.”  to Assumption A5 to clarify what a projection followed by a ReLU network means. For brevity, the fact that the target audience will be familiar with NN structure and since the reader can refer to Labate et al.’s work for more strict definitions (as stated at the start of section 4.1.1), we omit a mathematically strict definition.
>
> •	“The definition of “argmax” in Theorem 2 is unclear. For example, if the vector is (1,1,1) what is “the index” that the argmax gives?”
>
> Thank you for catching this issue in the definition of argmax. To make the exact behavior clear to the reader, I added a sentence describing how this is handled in argmax: .’If multiple entries attain the maximum, the first such index is returned’; consider e.g. numpy.argmax — NumPy v2.4 Manual, Softmax & Argmax | Kaggle
> While clear definitions are important, especially in a work as theory heavy as ours, most readers that conduct NN research will be familiar with the argmax function. Further, the described situation (1,1,1) is not of much interest in this case considering the Theorem’s statement since we claim that a separating hyperplane can be found which would not run into such a case.
>
> •	“In Definition 2, I guess the x and x are used interchangeably.”, “In Assumption A4, I guess the  and R are used interchangeably”
>
> These are correct. Thank you for catching these typing error!
>
> •	“The definition of  de in Assumption A5 is unclear.”
>
> Thank you for pointing out this concern regarding the definition of d_e. I added ‘The effective dimension d_e together with A and c are chosen in such a way that he projection fulfills the following properties’. Together with the already existing “Labate et al. show in their Theorem 4 (which we will call Theorem L4 to avoid confusion with Theorem 4 in this work) that this assumption is e.g. fulfilled if the following assumptions are met:”, it should be clear that a) and b) are example assumptions that fulfill the target assumption A5.
> I hope that this solves the issue. If not, please describe why d_e is not defined clearly.
>
> •	“Moreover, since A is a  d*D matrix and c a vector in Rd, it seems that d=de must hold. But then this contradicts Theorem L4 (A5 (b))”
>
> Again, thank you so much for pointing out this error! I corrected the statement to read that A is a  d_e*D matrix and c is a vector in R^d_e.
>
> •	“The assumptions of Theorem L4 and thus Theorem 4 are not applicable for compact flat manifolds, while Theorem 2 assumes compact flat manifolds. I wonder how they are related to one another?“
>
> Thank you for catching and pointing out this inconsistency. The reason why Theorem 2 only considers compact flat manifolds is that any two non-parallel affine linear subspaces meet at some point. We do not want this to happen as per the theorem’s assumption. Since the two cases (regression and classification) are considered inherently different in what the network should achieve, we did not consider how the theorems are related. However, whether a unifying view can be found would be an interesting question for future research.
>
> •	“In Remark 5,  is referred to as “projection dimension” instead of effective dimension”
>
> Thank you for pointing out that this switch in wording could confuse the reader. I chose this wording to stress in the context of the remark that d_e is the dimension that the projection’s output has. To avoid confusion, I changed the wording to ‘the effective dimension d_e (which is the dimension after the projection)’ .
>
> •	“In Corollary 1, an affine linear subspace is unbounded, thus cannot be contained in the cube.”
>
> Thank you for catching this inconsistency! I changed the formulation to “is a flat manifold as described in Theorem 1”, so the manifold is now bounded.

---

> > ### Author Response · Authors · 2026-01-14
> > **Answer Part 2**
> >
> > •	“In Corollary 2, M  is required to be a compact manifold in [0 1/2D]^D. When D is large (which is the case when low-dimensional structure of data becomes interesting), this is a small set, and may not represent the real case”
> >
> > I appreciate you voicing this concern. I added an remark at the end of the theory section (4.1) addressing this: “Note that the theorems and corollaries presented in this section have strict assumptions which are not fulfilled in many practical applications of NNs. Examples are Assumption A 5 about the structure of the considered NN, Assumption A 4 about the projection before the ReLU network, or the assumption from Corollaries 1 and 2 that M ⊂ [0, 1/2√D ]D which is only a small interval for large D. To check whether flat manifolds are still beneficial if these assumptions are not fulfilled, we conducted empirical experiments in the following Section 4.2.
> >
> > •	“There are other unclear things. Because of such confusion, I am unable to check the proofs and other technical things.”
> >
> > Thank you for pointing this out. We hope that the explanations in this answer address all the issues. If not, please elaborate on what information is still missing.
> >
> > •	“The experiments in Section 4.2.1 use Leaky ReLU instead of ReLU, thus cannot back the theoretical results”
> >
> > Thank you for bringing up this concern about the difference between experiments and theory. While you are correct that the experiments are different from the theoretical assumptions, this is exactly the point we wanted to make. We want to consider NN architectures which are closer to the ones used in practice than the theoretical assumptions allow. One difference is the lack of designated projection layer according to Assumption A5 and another difference is the use of the leaky ReLU activation function. For a positive real number as argument, this is the same as the ReLU activation function. However, negative arguments are not mapped to 0 but according to a linear function with small negative slope. This helps in computing meaningful parameter updates in the iterative gradient-based optimization process. We deemed that this shift from the theoretical assumptions towards practical applications by using a slight variation of the activation function would support the point that the result do not necessarily only hold for unrealistic assumptions.
> >
> > •	“The paper aims at showing the decrease of upper bound of the generalization error. Unless the generalization is of the same order as the upper bound, the generalization error may not change at all! Indeed, the paper proposes an interesting point of view, but it is not enough to explain what the title suggests, namely generalization would be better for flatter manifolds.”
> >
> > Thank you for pointing out this concern.
> > This is correct. The title “On the generalization superiority of flat representation manifolds in deep learning machines” does not state that flat manifolds result in better generalization, but only that this topic is considered in the paper and the question is considered whether, how and in what context flat representation manifolds are beneficial for generalization. In the abstract and the rest of the paper we state clearly, that the upper bound on the generalization gap is decreases, not the generalization gap itself. Furthermore, we openly state in Remark 4 that the upper bound is too large to have meaning in practice. This flaw is inherited from Labate et al.’s work. We consider the almost-flat case in more detail since this provides better upper bounds. It is general practice in the field of machine learning that considers generalization to work with upper bound on the generalization gap (see e.g. VC-dimension or Rademacher complexity). These also do not guarantee generalization performance,  but only provide an upper bound.
> > In the conclusion we state that an improvement in Theorem L4 would result in a better bound, so one way to improve the bound is through improving this theorem.

---

> ### Author Response · Authors · 2026-01-14
> **Answer Part 3**
>
> •	“It would be better if you could strengthen the results.”
>
> I appreciate you pointing out this concern about the amount of progress that the paper presents.
> Gaining theoretical insights into NNs is a complex task and as such, in our experience, needs many small steps to achieve groundbreaking results. Basing results upon prior research by other researchers makes sure that their interesting work does not go unnoticed and reaches its full potential. Even though many of our results are built upon prior research, they are nonetheless interesting results on their own. From Labate et al.’s work, you still need many computations to prove that the monotonicity holds. This means that other researchers considering this work do not have to go through developing the strenuous computations themselves to reach the result we already found. Further, the insights into flat and almost-flat manifolds are interesting steps to further understanding of NN behavior.
> However, we agree that this line of research has a lot more potential and is not done yet. For example, improving Labate et al.’s Theorem L4 would improve the upper bound on the generalization gap massively. With our proof of Theorem 4, it would be relatively easy to check whether the monotonicity still holds if a better bound is found.
> This will be goals for future research. However, as stated prior, we think that the existing results are already interesting, thorough and far-reaching enough to be of interest on their own as basis for future understanding of NN behaviour. Further, they provide grounds on which the research community can build upon.
> To stress this promising research direction, we added the following to the outlook: “One major point of interest for this is proving that Assumption A 5 can be fulfilled with smaller effective dimension de in the general case (not only the flat or almost-flat case). This will be grounds for future research.”
>
> Thank you again so much for reading the manuscript so thoroughly and pointing out various concerns, comments and suggestions. I hope the answers provided help clear up some of your concerns.
> I’m looking forward to your reply!

---

> ### Author Response · Authors · 2026-01-14
> **Changes in Manuscript**
>
> Changes in the manuscript due to first review round:
> •	Abstract
> o	I changed the wording from ‘perfect classification’ to ‘perfect class separability’ in the following sentence: “Under the manifold hypothesis, we
> demonstrate that flat manifolds (subsets of affine linear subspaces) in the second-to-last layer of a classification network ensure perfect class separability in the noiseless case.”
>
> •	Section 1 Introduction
> o	I added the following as one main contribution at the end of the introduction “A theorem showing that flat class manifolds in the second-to-last layer of a classification network ensure perfect class separability in the noiseless case (Theorem 2)”
>
> •	Section 2
> o	 I corrected a typo in the definition of the reach (Definition 2) where previously one of the x was not in math mode.
>
> •	Section 3
> o	I added ’If multiple entries attain the maximum, the first such index is returned’ to Theorem 2 to define argmax in more detail.
>
> o	I added the following sentences to Remark 3 to argue that flat representation manifolds can be achieved by NN transformations: “Note that Berner et al. (2023) argued in their Section 4.1 why NNs can, in the presence of a data manifold, reduce the problem to the underlying low-dimensional problem. This is done by partitioning the manifold into suitable neighborhoods and approximating the coordinate charts of the manifold via NNs using the universal approximation property. This means that NNs can recover the low-dimensional manifold-inherent coordinate system. If they can perform this, they can also project the resulting low-dimensional manifold into high-dimensional space, resulting in flat manifolds.
> Further, Theorem 2 does not state how to achieve flat manifolds in the second-to-last layer. One idea that we want to consider in future research is to encourage flat representation manifolds by explicitly measuring the flatness and adding this as penalty term in the loss function, resulting in a flatness regularizer.”
>
> •	Section 4.1
> o	In Assumption A5 I added the following to more clearly define de: ‘The effective dimension d_e together with A and c are chosen in such a way that he projection fulfills the following properties’
>
> o	I corrected in Assumption 5 that A is a d_e*D matrix and c is a d_e vector.
>
>
> o	I added to Assumption A6: “A ReLU network is a fully connected feed forward NN where the activation function is the well-known rectified linear unit function.” to explain ReLU networks. Further, I added “maximal norm of the weights B ∈ R.” to clarify the role of B and  “This means that the output of the projection is the input for the ReLU NN further detailed in Assumption A 6.”  to explain how the projection and ReLU networks are combined.
>
> o	I changed the wording in Remark 5 to be ‘the effective dimension d_e (which is the dimension after the projection)’ to consistently use the wording ‘effective dimension’ for d_e.
>
>
> o	In Corollary 1 I corrected a mistake in the formulation, changing the wording to “is a flat manifold as described in Theorem 1”.
>
> o	I added Remark 8 to better motivate why we need the empirical experiments
>
>
> •	Section 4.2
> o	I corrected a mistake in computing the base estimator’s test error. This was now lowered to 0.7 or 0.6. The points made by the experiments stay the same.
>
> •	Section 5 Discussion
>
> •	Section 6 Outlook
> o	I added “One major point of interest for this is proving that Assumption A 5 can be fulfilled with smaller effective dimension de in the general case (not only the flat or almost-flat case). This will be grounds for future research.”

---

### Review · Reviewer_5kAa · 2025-12-23

**Summary Of Contributions:**

1. This paper studies the relationship between the flatness of data and representation manifolds and the generalization performance of deep neural networks, under the manifold hypothesis. The authors investigate both classification and regression settings.

2. For classification, the paper shows that when class-specific representation manifolds in the penultimate layer are flat (i.e., subsets of affine subspaces) and non-intersecting, they are linearly separable, guaranteeing perfect classification in the noiseless case.

3. For regression, building on recent generalization bounds for neural networks trained on data supported on manifolds, the paper analyzes how the curvature of the input manifold—quantified through the reach and related geometric notions—affects the practical dimension appearing in the bound. The authors show that increasing flatness can reduce this practical dimension and thus lower the resulting upper bound on the generalization error. Furthermore, they derive refined results for almost-flat manifolds, showing that under suitable deviation conditions, the practical dimension can be reduced to the manifold's intrinsic dimension. Numerical experiments illustrate that flatter manifolds can empirically improve generalization, even beyond the strict theoretical assumptions.

**Additional Comments:**

This paper tackles an important and timely question with careful mathematical reasoning. While the current theoretical results may fall short of the strongest standards expected for a top-tier theory venue, the work represents a thoughtful and serious attempt to bridge geometric intuition and generalization theory. With a sharper theoretical focus and clearer conceptual framing, the ideas developed here could form the basis of a strong future contribution.

**Audience:**

Yes

**Audience Explanation:**

The topic addressed in this paper, understanding generalization in deep learning through the geometry of data and representation manifold, is of clear interest to the theoretical machine learning community. Researchers working on the manifold hypothesis, representation learning, implicit regularization, and geometric perspectives on deep networks would likely find the questions studied in this paper relevant. In particular, the analysis of almost-flat manifolds and the explicit decomposition of representations into linear and residual components touches on a central theme in modern representation learning: the emergence of approximately linear structures in deep networks. Even though loose bounds limit the current theoretical results, the paper contributes to an active line of inquiry that remains far from settled.

**Broader Impact Concerns:**

The paper is primarily theoretical and does not raise immediate ethical or societal concerns. Its focus on understanding generalization in deep learning could indirectly contribute to the development of more reliable and efficient models, which is generally positive. No negative broader impacts are apparent.

**Claims And Evidence:**

Yes

**Claims Explanation:**

The claims made in the paper are generally mathematically correct and supported by sound reasoning under the stated assumptions. In particular, the results on classification separability and the analysis of almost-flat manifolds are internally consistent and well justified. The empirical results are consistent with the theoretical intuition but are limited in scope, relying on synthetic manifolds and relatively small performance gaps. As such, they provide supporting evidence but are not fully convincing on their own.

**Requested Changes:**

1. Clarify and unify the notion of flatness. The paper would benefit from a clearer conceptual discussion that distinguishes and relates the different concepts of flatness used (second fundamental form, reach, and residual deviations). Ideally, the authors should identify which notion is most relevant for generalization and why.

2. Strengthen the theoretical contribution beyond monotonicity. The current main regression result primarily establishes a monotonic dependence of an existing bound on an effective dimension parameter. Providing sharper bounds, alternative complexity measures, or new theoretical insights not directly inherited from prior work would significantly strengthen the contribution.

3. Elevate the role of almost-flat manifolds. The results for almost-flat manifolds appear to be among the most novel and conceptually meaningful parts of the paper. Reframing these results as central rather than peripheral could improve the overall impact.

4. Expand or deepen the empirical evaluation. Additional experiments, such as more diverse manifold constructions, larger-scale settings, or analyses that directly probe effective dimensionality, would help support the theoretical claims and improve confidence in their practical relevance.

---

> ### Author Response · Authors · 2026-01-14
> **Answer Reviewer 5kAa Part 1**
>
> Dear Reviewer,
> I appreciate your thorough and valuable feedback and the opportunity to address your comments, suggestions and concerns.
>
> •	“Clarify and unify the notion of flatness. The paper would benefit from a clearer conceptual discussion that distinguishes and relates the different concepts of flatness used (second fundamental form, reach, and residual deviations). Ideally, the authors should identify which notion is most relevant for generalization and why.”
>
> Thank you for pointing out the fact that the different flatness notions could be confusing. Remark 1 describes the connection between reach and SFF. SFF is more common to use in differential geometry which is why I started with this notion. The reach is then a combination of SFF and the smallest bottleneck (as described in Remark 1). The residual deviations are different from the reach and SFF in the sense that they are no local measure, but the differences between all t_x and t_y and corresponding x and y are considered. The SFF cannot be decomposed into the part assigned to the residual deviation and the linear part. As such, after some consideration, I don’t think that a unifying view can be established, at least between reach/SFF and the residual deviation. I understand that readers would benefit from more clarity, but I am afraid unifying the three notions is not possible in the general case. Reach and residual deviation are, depending on the considered case, most important for generalization according to our theoretical analysis.
>
> •	“Strengthen the theoretical contribution beyond monotonicity. The current main regression result primarily establishes a monotonic dependence of an existing bound on an effective dimension parameter. Providing sharper bounds, alternative complexity measures, or new theoretical insights not directly inherited from prior work would significantly strengthen the contribution.”
> I appreciate this concern regarding the amount of progress presented in the paper. Gaining theoretical insights into NNs is a complex task and as such, in our experience, needs many small steps to achieve groundbreaking results. Basing results upon prior research by other researchers makes sure that their interesting work does not go unnoticed and reaches its full potential. Even though many of our results are built upon prior research, they are nonetheless interesting results on their own. From Labate et al.’s work, you still need many computations to prove that the monotonicity holds. This means that other researchers considering this work do not have to go through developing the strenuous computations themselves to reach the result we already found. Further, the insights into flat and almost-flat manifolds are interesting steps to further understanding of NN behavior.
> However, we agree that this line of research has a lot more potential and is not done yet. For example, improving Labate et al.’s Theorem L4 would improve the upper bound on the generalization gap massively. With our proof of Theorem 4, it would be relatively easy to check whether the monotonicity still holds if a better bound is found.
> These will be goals for future research. However, as stated prior, we think that the existing results are already interesting, thorough and far-reaching enough to be of interest on their own as basis for future understanding of NN behavior. Further, they provide grounds which the research community can build upon.
> To stress this promising research direction, we added the following to the outlook: “One major point of interest for this is proving that Assumption A 5 can be fulfilled with smaller effective dimension de in the general case (not only the flat or almost-flat case). This will be grounds for future research.”

---

> > ### Author Response · Authors · 2026-01-14
> > **Answer Reviewer 5kAa Part 2**
> >
> > •	“Elevate the role of almost-flat manifolds. The results for almost-flat manifolds appear to be among the most novel and conceptually meaningful parts of the paper. Reframing these results as central rather than peripheral could improve the overall impact.”
> >
> > Thank you for this suggestion. I understand that the result for almost-flat manifolds is the most novel part in terms of our own considerations. However, most data manifolds in practice won’t be almost-flat. Representation manifolds in the last layers of a NN can be of the almost-flat type, which is why this is still an interesting result. However, the results for non-almost-flat manifolds are still very meaningful. They are also needed since after training the representation manifolds in the first layers will not be almost-flat due to their proximity to the raw data (input layer). Further, continuing to develop other researcher’s prior work is a major reason for publishing one’s own work. Even though my Theorem 4 is based on prior work, I think that it still has an important role to play, especially since it considers many manifolds which are not almost-flat.
> >
> > •	“Expand or deepen the empirical evaluation. Additional experiments, such as more diverse manifold constructions, larger-scale settings, or analyses that directly probe effective dimensionality, would help support the theoretical claims and improve confidence in their practical relevance.”
> >
> > We appreciate your suggestions for strengthening the empirical evaluations in the manuscript. We want the focus of the manuscript to be on the theoretical results. The empirical experiments are meant to serve as a first indication that the gap between theory and practical applications may not be as large as indicated by our theoretical results.
> > However, in future work we aim to conduct more experiments, e.g. having more diverse manifolds such as ones generated using polynomial curvatures, having multiple curves or spirals.
> > Regarding the suggestion of probing the effective dimension, we conducted experiments that optimize the loss of information caused by the projection by choosing an optimal projection matrix via nonlinear optimization. This optimization is a lengthy process, however we found that even d_e=1 worked well for flat manifolds while even d_e=2d did not work for curvy manifolds. We see this as more of a step in between the theory and our conducted experiments since the fact that flat manifolds can be modelled better than curvy manifolds by NNs is built upon the assumptions that d_e can be lower for flat manifolds. Because of this, and the already lengthy manuscript which focuses on theory rather than empirical experiments, we would rather not add this experiment to the manuscript.
> >
> > Again, we want to thank you for engaging so thoroughly with our work and its presentation. We hope that the arguments presented here can alleviate some of your concerns and answer some of your questions.
> > I’m looking forward to your reply!

---

> > > ### Author Response · Authors · 2026-01-14
> > > **Changes in Manuscript**
> > >
> > > Changes in the manuscript due to first review round:
> > > •	Abstract
> > > o	I changed the wording from ‘perfect classification’ to ‘perfect class separability’ in the following sentence: “Under the manifold hypothesis, we
> > > demonstrate that flat manifolds (subsets of affine linear subspaces) in the second-to-last layer of a classification network ensure perfect class separability in the noiseless case.”
> > >
> > > •	Section 1 Introduction
> > > o	I added the following as one main contribution at the end of the introduction “A theorem showing that flat class manifolds in the second-to-last layer of a classification network ensure perfect class separability in the noiseless case (Theorem 2)”
> > >
> > > •	Section 2
> > > o	 I corrected a typo in the definition of the reach (Definition 2) where previously one of the x was not in math mode.
> > >
> > > •	Section 3
> > > o	I added ’If multiple entries attain the maximum, the first such index is returned’ to Theorem 2 to define argmax in more detail.
> > >
> > > o	I added the following sentences to Remark 3 to argue that flat representation manifolds can be achieved by NN transformations: “Note that Berner et al. (2023) argued in their Section 4.1 why NNs can, in the presence of a data manifold, reduce the problem to the underlying low-dimensional problem. This is done by partitioning the manifold into suitable neighborhoods and approximating the coordinate charts of the manifold via NNs using the universal approximation property. This means that NNs can recover the low-dimensional manifold-inherent coordinate system. If they can perform this, they can also project the resulting low-dimensional manifold into high-dimensional space, resulting in flat manifolds.
> > > Further, Theorem 2 does not state how to achieve flat manifolds in the second-to-last layer. One idea that we want to consider in future research is to encourage flat representation manifolds by explicitly measuring the flatness and adding this as penalty term in the loss function, resulting in a flatness regularizer.”
> > >
> > > •	Section 4.1
> > > o	In Assumption A5 I added the following to more clearly define de: ‘The effective dimension d_e together with A and c are chosen in such a way that he projection fulfills the following properties’
> > >
> > > o	I corrected in Assumption 5 that A is a d_e*D matrix and c is a d_e vector.
> > >
> > >
> > > o	I added to Assumption A6: “A ReLU network is a fully connected feed forward NN where the activation function is the well-known rectified linear unit function.” to explain ReLU networks. Further, I added “maximal norm of the weights B ∈ R.” to clarify the role of B and  “This means that the output of the projection is the input for the ReLU NN further detailed in Assumption A 6.”  to explain how the projection and ReLU networks are combined.
> > >
> > > o	I changed the wording in Remark 5 to be ‘the effective dimension d_e (which is the dimension after the projection)’ to consistently use the wording ‘effective dimension’ for d_e.
> > >
> > >
> > > o	In Corollary 1 I corrected a mistake in the formulation, changing the wording to “is a flat manifold as described in Theorem 1”.
> > >
> > > o	I added Remark 8 to better motivate why we need the empirical experiments
> > >
> > >
> > > •	Section 4.2
> > > o	I corrected a mistake in computing the base estimator’s test error. This was now lowered to 0.7 or 0.6. The points made by the experiments stay the same.
> > >
> > > •	Section 5 Discussion
> > >
> > > •	Section 6 Outlook
> > > o	I added “One major point of interest for this is proving that Assumption A 5 can be fulfilled with smaller effective dimension de in the general case (not only the flat or almost-flat case). This will be grounds for future research.”

---

### Review · Reviewer_cLEF · 2026-01-06

**Summary Of Contributions:**

This paper tackles the generalisation ability of Neural Networks by investigating the manifold hypothesis. This hypothesis states that the input data lie in a low-dimensional manifold, and that NNs have the ability to learn a representation manifold. More precisely, the authors mainly claim the following contribution:
1) In the regression case, they show that flatter input manifolds lead to tighter upper bounds.
2) They provide synthetic experiments to confirm the benefits of flat input manifolds in regression empirically.

Strengths:
1) The paper is well motivated.
2) The theoretical claims are supported by proofs.

Weaknesses:
1) The paper strongly lacks clarity. In particular, the authors claim a contribution in the classification settings in the abstract, while it is missing in the end of the Introduction section.
2) Section 3 about classification seems superficial. Theorem 2 shows that two flat manifolds that do not intersect can be separated by a linear NN, however there is no result on the ability of the NN to learn such manifold representations from input data.
3) The organisation of the paper is unbalanced. Section 3 should be focused on the theoretical analysis of the regression case (current section 4.1) and Section 4 on the empirical results (current section 4.2).
4) The theoretical findings are very incremental.
5) The empirical analysis is not very convincing.

**Audience:**

No

**Audience Explanation:**

Investigating the manifold hypothesis is indeed of interest, however I do not think that the authors achieved providing substantial findings.

Regarding the classification case, Theorem 2 seems superficial and does not match the initial claim in the abstract.

Regarding the regression case, the main result consists in proving that the generalisation bound provided in Labate & Shi (2024) is indeed decreasing in $d_e$, showing that the bound gets tighter with less curvy input manifold. Moreover, the authors prove a Corollary that holds with almost flat manifolds. These results are nice but are fairly incremental.

Finally, The empirical analysis is not very convincing, as detailed in the previous section of the review.

**Claims And Evidence:**

No

**Claims Explanation:**

All the theoretical results are supported with proofs. However, a few points need to be tackled.

First, the following claim in the abstract "Under the manifold hypothesis, we demonstrate that flat manifolds (affine linear subspaces) in the second-to-last layer of a classification network ensure perfect classification performance in the noiseless case" is not supported by an evidence. Theorem 2 shows that two flat manifolds that do not intersect can be separated by a linear NN, however there is no result on the ability of the NN to learn such manifold representations from input data.

Moreover, the empirical analysis is not very convincing. No experiments on real-world data are provided. The experiment settings consist in comparing the results obtained with two different NN architectures on two different input manifolds (less and more curvy). It is difficult to evaluate to what extent the difference between the test errors is substantial. Finally, according to the displayed results, adding an L1-regulariwer seems to increase the error, which is contradictory to Remark 6.

**Requested Changes:**

The authors should address my concerns exposed in the previous sections of the review, even though I do not think that a minor revision would be sufficient to do so.

---

> ### Author Response · Authors · 2026-01-14
> **Answer to reviewer cLEF Part 1**
>
> Dear Reviewer,
> Thank you so much for your thorough review of our manuscript. I appreciate the time and effort you have dedicated to reviewing this work. Below, I address your comments and suggestions:
> •	„the authors claim a contribution in the classification settings in the abstract, while it is missing in the end of the Introduction section.”
>
> Thank you for highlighting this inconsistency. I added this as one item in the main contributions “A theorem showing that flat class manifolds in the second-to-last layer of a classification network ensure perfect class separability in the noiseless case (Theorem 2)”.
>
> •	“Theorem 2 shows that two flat manifolds that do not intersect can be separated by a linear NN, however there is no result on the ability of the NN to learn such manifold representations from input data.[…] the following claim in the abstract "Under the manifold hypothesis, we demonstrate that flat manifolds (affine linear subspaces) in the second-to-last layer of a classification network ensure perfect classification performance in the noiseless case" is not supported by an evidence. Theorem 2 shows that two flat manifolds that do not intersect can be separated by a linear NN, however there is no result on the ability of the NN to learn such manifold representations from input data.”
>
> Thank you for pointing out this question readers could have after reading the classification section of the manuscript.
> Your statement is correct. However, we never claimed that a NN would be able to reach such representations. Only that if there are flat representations in the second-to-last layers, then a linear NN layer can be found that reaches perfect classification.
>
> As to the question, whether finding such representations is possible, we added the following to Remark 3: “Note that Berner et al. (2023) argued in their Section 4.1 why NNs can, in the presence of a data manifold, reduce the problem to the underlying low-dimensional problem. This is done by partitioning the manifold into suitable neighborhoods and approximating the coordinate charts of the manifold via NNs using the universal approximation property. This means that NNs can recover the low-dimensional manifold-inherent coordinate system. If they can perform this, they can also project the resulting low-dimensional manifold into high-dimensional space, resulting in flat manifolds.”
> To clarify that finding such suitable parameters is still a problem, we added the following to Remark 3: “Theorem 2 does not state how to achieve flat manifolds in the second-to-last layer. One idea that we want to consider in future research is to encourage flat representation manifolds by explicitly measuring the flatness and adding this as penalty term in the loss function, resulting in a flatness regularizer.”’  In addition, we changed the wording in abstract to ‘ensures perfect separability’, highlighting that perfect classification can be achieved in such a case by a linear layer if suitable parameters are found.  Whether gradient descent can efficiently find the required NN parameters (maybe in combination with the described regularizer), will be an interesting topic for future research.
>
> •	“The organisation of the paper is unbalanced.  Section 3 should be focused on the theoretical analysis of the regression case (current section 4.1) and Section 4 on the empirical results (current section 4.2).”
>
> I appreciate your feedback on the outline of the manuscript. I’m afraid I do not know what you mean by ‘The organisation of the paper is unbalanced.‘ I designed the structure to first consider the classification case, then the regression case since these two cases have to be considered separately. Since the theorems in the regression case contain many unrealistic assumptions (in practical applications), the empirical section was required to argue why flat manifolds can still be useful in real applications. Such an empirical argument is not required in the classification case. I hope this explains why the sections are structured the way they are. If the explanations did not satisfy you in this regard, please elaborate on the reason for changing the sections.

---

> > ### Author Response · Authors · 2026-01-14
> > **Answer to reviewer cLEF Part 2**
> >
> > •	The theoretical findings are very incremental.
> >
> > Thanks for sharing your concerns regarding the amount of progress this paper presents.
> > As explained in the introduction, theoretical analysis of the generalization performance of neural networks in combination with the manifold hypothesis has only been described in few (mainly 1) article(s). In general, theoretical analysis of NNs is only sparsely available and more often than not conducted in small steps due to its complexity.  As such, small but consistent steps are needed to further this field in NN research. Further research, based on these initial results could provide more groundbreaking statements. However, publishing these results would already be beneficial as it could encourage the research community to work in this direction, which we find promising. We will continue to work on this issue in the future, so we hope that this paper presents a first step to more groundbreaking results down the line.
> >
> > •	The empirical analysis is not very convincing: No experiments on real-world data are provided.
> >
> > I am grateful for your observations regarding the experimental section. We plan to consider real-world data in the future. The goal of the experimental section here was to support the claim that the theorems’ statements could also hold for cases which are closer to practical applications. As such, we need manifolds which have different  curvatures but are the same in all other aspects, e.g. general shape. This way, we can definitely point to the flatness as the cause of the difference in results regarding both manifolds. When considering real-world data, we would not have such a definite comparison. Since the paper has already a length of over 12 pages and the focus should be on the theoretical results, we did not want to add real-world data. However, applying our findings to benefit real-world applications with real-world data is the focus of our ongoing work.
> >
> > •	“The experiment settings consist in comparing the results obtained with two different NN architectures on two different input manifolds (less and more curvy). It is difficult to evaluate to what extent the difference between the test errors is substantial.”
> >
> > Thank you for pointing out this question. To show that the difference is statistically significant, we used the p-value of a t-test. The result states that the difference in test error is (with high probability) not due to chance alone, but that the curviness of the manifold is important. Our aim was to show that there is an advantage in having flat manifolds, not quantify that advantage since this depends on the data, architecture and so on. Since we consider simulated data (the reason for this explained above), quantifying the benefit of flat manifolds would not be of much interest.
> >
> > •	“Finally, according to the displayed results, adding an L1-regulariwer seems to increase the error, which is contradictory to Remark 6.”
> >
> > Thank you so much for pointing out this inconsistency. The point that we wanted to make with the experiment, i.e. that flat manifolds are beneficial for generalization when combining them with L1 regularizers, still stands, whether the results with or without regularizer perform better overall. However, this is a sign that the regularization causes a loss of information. The reason for this is that the noise level (signal-to-noise ratio of 70) was too low compared to the signal to cause the network without regularization to overfit. We did not perform an optimization to find the best weight for the L1 regularizer, so the one we chose might be too high, causing the loss of information. Nevertheless, the point we wanted to make, i.e. that the L1 regularizer is needed in this setting to differentiate between the flat and curvy manifold in terms of their performance on a test set still stands. To harness the benefits of flat manifolds in settings with large networks, the L1 regularizer has to be added.
> >
> > I hope these answers help clarify some of your comments, questions and concerns. Again, we thank you wholeheartedly for engaging in such a thorough way with our manuscript and research and helping us improve our work.
> > I’m looking forward to your reply!

---

> > > ### Author Response · Authors · 2026-01-14
> > > **Changes in Manuscript**
> > >
> > > Changes in the manuscript due to first Review round:
> > > •	Abstract
> > > o	I changed the wording from ‘perfect classification’ to ‘perfect class separability’ in the following sentence: “Under the manifold hypothesis, we
> > > demonstrate that flat manifolds (subsets of affine linear subspaces) in the second-to-last layer of a classification network ensure perfect class separability in the noiseless case.”
> > >
> > > •	Section 1 Introduction
> > > o	I added the following as one main contribution at the end of the introduction “A theorem showing that flat class manifolds in the second-to-last layer of a classification network ensure perfect class separability in the noiseless case (Theorem 2)”
> > >
> > > •	Section 2
> > > o	 I corrected a typo in the definition of the reach (Definition 2) where previously one of the x was not in math mode.
> > >
> > > •	Section 3
> > > o	I added ’If multiple entries attain the maximum, the first such index is returned’ to Theorem 2 to define argmax in more detail.
> > >
> > > o	I added the following sentences to Remark 3 to argue that flat representation manifolds can be achieved by NN transformations: “Note that Berner et al. (2023) argued in their Section 4.1 why NNs can, in the presence of a data manifold, reduce the problem to the underlying low-dimensional problem. This is done by partitioning the manifold into suitable neighborhoods and approximating the coordinate charts of the manifold via NNs using the universal approximation property. This means that NNs can recover the low-dimensional manifold-inherent coordinate system. If they can perform this, they can also project the resulting low-dimensional manifold into high-dimensional space, resulting in flat manifolds.
> > > Further, Theorem 2 does not state how to achieve flat manifolds in the second-to-last layer. One idea that we want to consider in future research is to encourage flat representation manifolds by explicitly measuring the flatness and adding this as penalty term in the loss function, resulting in a flatness regularizer.”
> > >
> > > •	Section 4.1
> > > o	In Assumption A5 I added the following to more clearly define de: ‘The effective dimension d_e together with A and c are chosen in such a way that he projection fulfills the following properties’
> > >
> > > o	I corrected in Assumption 5 that A is a d_e*D matrix and c is a d_e vector.
> > >
> > >
> > > o	I added to Assumption A6: “A ReLU network is a fully connected feed forward NN where the activation function is the well-known rectified linear unit function.” to explain ReLU networks. Further, I added “maximal norm of the weights B ∈ R.” to clarify the role of B and  “This means that the output of the projection is the input for the ReLU NN further detailed in Assumption A 6.”  to explain how the projection and ReLU networks are combined.
> > >
> > > o	I changed the wording in Remark 5 to be ‘the effective dimension d_e (which is the dimension after the projection)’ to consistently use the wording ‘effective dimension’ for d_e.
> > >
> > >
> > > o	In Corollary 1 I corrected a mistake in the formulation, changing the wording to “is a flat manifold as described in Theorem 1”.
> > >
> > > o	I added Remark 8 to better motivate why we need the empirical experiments
> > >
> > >
> > > •	Section 4.2
> > > o	I corrected a mistake in computing the base estimator’s test error. This was now lowered to 0.7 or 0.6. The points made by the experiments stay the same.
> > >
> > > •	Section 5 Discussion
> > >
> > > •	Section 6 Outlook
> > > o	I added “One major point of interest for this is proving that Assumption A 5 can be fulfilled with smaller effective dimension de in the general case (not only the flat or almost-flat case). This will be grounds for future research.”

---

### Decision · Action_Editor_Grac · 2026-03-12

**Recommendation:** Reject

**Additional Comments:**

Reviewers remain concerned about the overall strength and scope of the contributions in their current form: while the analysis appears technically sound, the main theoretical results are deemed too incremental with respect to existing bounds and the empirical evaluation remains limited. Although TMLR values technical correctness even for modest contributions, the combination of these concerns and the overall negative reviewer assessments leads me to recommend rejection at this stage, with the possibility of resubmitting a major revision if the authors wish to do so.

**Audience:**

No

**Audience Explanation:**

The reviewers agree that the paper studies a question that is relevant to the TMLR audience, particularly at the intersection of manifold geometry and generalisation in deep learning, but reviewers consider the paper's contributions too thin to trigger interest.

**Claims And Evidence:**

Yes

**Claims Explanation:**

The reviewers generally agree that the paper is mathematically careful.

**Resubmission Of Major Revision:**

The authors may consider submitting a major revision at a later time.